# Identification of the PTEN-ARID4B-PI3K pathway reveals the dependency on ARID4B by PTEN-deficient prostate cancer

Ray-Chang Wu[1], In-Chi Young[1], Yu-Fang Chen[1], Sung-Ting Chuang[1], Antoun Toubaji[2] & Mei-Yi Wu[1,3]

*PTEN* is frequently mutated in prostate cancer. The tumor suppressor function of PTEN is attributed to its lipid phosphatase activity that counters PI3K action. Here, we report a PTEN-ARID4B-PI3K axis in which PTEN inhibits expression of *ARID4B*, while ARID4B is a transcriptional activator of the PI3K subunit genes *PIK3CA* and *PIK3R2* that are crucial for activation of the PI3K/AKT pathway. Reciprocal binding of ARID4B and histone H1 to the *PIK3CA* and *PIK3R2* promoters modulates chromatin condensation, suggesting a mechanism by which ARID4B activates these promoters. Functional analyses reveals that ARID4B is required for prostate tumorigenesis when PTEN is deficient. The biological significance is further substantiated by the existence of a *PTEN/ARID4B/PIK3CA* three-gene signature that improves the predictive power for prostate cancer recurrence in patients. In summary, we identify ARID4B as a master regulator in the PTEN-PI3K pathway, thus providing a potential therapeutic target for prostate cancer carrying *PTEN* mutations.

[1] Department of Biochemistry and Molecular Medicine, The George Washington University, Washington, DC 20037, USA. [2] Department of Pathology, The George Washington University, Washington, DC 20037, USA. [3] Department of Medicine, The George Washington University, Washington, DC 20037, USA. Correspondence and requests for materials should be addressed to R.-C.W. (email: rwu@gwu.edu) or to M.-Y.W. (email: meiyiwu@gwu.edu)

Prostate cancer is the second leading cause of male cancer-related death worldwide. Loss of the tumor suppressor gene *PTEN* is a frequent genetic alteration in prostate cancers, and many late-stage prostate cancers showed altered activity in the PTEN-PI3K pathway[1,2]. To date, most targeted anticancer drugs inhibit oncogenes that acquire gain-of-function mutations in cancer. However, treatment of cancers that harbor loss-of-function mutations on tumor suppressor genes, such as *PTEN*, remains a significant challenge because there are no direct targets amenable to pharmaceutical intervention. The emerging concept of synthetic essentiality in cancer describes cancers with specific tumor suppressor deficiencies that depend on expression of synthetic-essential genes. Targeting such genes may be a promising alternative treatment approach for cancers that acquire loss-of-function mutations on tumor suppressor genes[3]. Therefore, significant efforts have been devoted to identifying synthetic-essential genes in PTEN-deficient prostate cancers[4,5].

PI3K phosphorylates phosphatidylinositol (PtdIns)-4,5-bisphosphate (PIP2) to generate PtdIns-3,4,5-trisphosphate (PIP3), which activates the AKT signaling pathway to mediate cellular events, including cell survival, growth, proliferation, and cell migration[6,7]. Hyperactivation of the PI3K pathway is associated with prostate cancer progression[8]. The biological effects of PTEN are dominated by its ability to dephosphorylate PIP3 back into PIP2, leading to suppression of the proto-oncogenic PI3K-AKT signaling pathway[6]. Three classes of PI3Ks (class I, class II, and class III) have been identified in mammals. Class I PI3Ks are heterodimeric enzymes that consist of a catalytic subunit p110α, p110β, p110γ, or p110δ (encoded by *PIK3CA*, *PIK3CB*, *PIK3CG*, or *PIK3CD*, respectively) and a regulatory subunit p85α, p55α, p50α (all encoded by *PIK3R1*), p85β (encoded by *PIK3R2*), p55γ (encoded by *PIK3R3*), p150 (encoded by *PIK3R4*), or p101 (encoded by *PIK3R5*)[8]. *PIK3CA* is frequently mutated in prostate cancers, and many of these mutations result in activation of PI3K[9]. In addition, the *Pik3ca* gain-of-function mutation accelerates cancer progression of PTEN-deficient prostate in mice[10]. Although alterations of the PTEN-PI3K-AKT pathway are strongly implicated in prostate cancer pathogenesis, their therapeutic targeting has not improved clinical efficacy[8]. Therefore, identification of effectors of this important pathway to target PTEN-deficient prostate cancers is urgently needed.

ARID4B belongs to the ARID family and is a chromatin remodeling factor[11]. ARID4B was previously identified as a component of the HDAC1/SIN3A chromatin remodeling complex[12]. Although ARID4B is a chromatin remodeling protein, it lacks methyltransferase or acetyltransferase activities. To date, how ARID4B modulates chromatin modifications to regulate gene expression is not well understood. However, ARID4B contains a Tudor domain and a chromo domain, and these domains may function as adaptors to assemble chromatin remodeling complexes[13,14]. Expression of ARID4B is elevated in many types of tumors[15–18], and ARID4B promotes mammary tumor growth and metastasis[19]. Despite these early findings, the role of ARID4B in prostate cancer remains unclear.

In the present study, we identify ARID4B as a regulator of the PTEN-PI3K pathway by functioning as a transcriptional activator of *PIK3CA* and *PIK3R2*. Importantly, we demonstrate that ARID4B is required for initiation and progression of PTEN-deficient prostate cancer, suggesting ARID4B is a potential therapeutic target for prostate cancer harboring *PTEN* mutations.

## Results

**ARID4B regulates the PI3K-AKT pathway**. To determine genomic alterations of *ARID4B* in prostate cancer, we analyzed the Cancer Genome Atlas (TCGA) and other prostate cancer gene expression

genomic datasets[2,20–25]. Amplification of *ARID4B* was detected in up to 20% of human prostate cancer cases (Supplementary Fig. 1a, left), while deletion and loss-of-function mutation of *PTEN* are frequent (Supplementary Fig. 1a, right). Using two prostate cancer gene expression datasets in the Oncomine database[23,26], we found that expression of *ARID4B* was significantly increased in prostate carcinoma (PCa) compared to normal prostate glands (Supplementary Fig. 1b). In addition, the human prostate cancer cell lines PC3, DU145, LNCaP, and C4-2B expressed higher levels of ARID4B protein than the normal (nonmalignant) prostate epithelial cell lines RWPE-1, PZ-HP-7, and HPrEC (Supplementary Table 1 and Supplementary Fig. 1c). Furthermore, patients expressing higher levels of *ARID4B* had a significantly increased risk of prostate-specific antigen recurrence in two datasets[27,28] (Supplementary Fig. 1d). These results indicate that expression of *ARID4B* positively correlates with prostate cancer development, and tumor recurrence is sensitive to higher levels of *ARID4B* expression.

To determine how ARID4B plays a role in prostate cancer, we performed whole transcriptome analysis (RNA-Seq) to compare gene expression profiles of PC3 cells with or without knockdown of *ARID4B* (Supplementary Data 1). Expression of the PI3K subunit genes *PIK3CA* and *PIK3R2* was significantly reduced in PC3 cells with knockdown of *ARID4B* (Table 1 and Supplementary Data 1). Independent quantitative RT-PCR (qRT-PCR) analysis confirmed decreased expression of *PIK3CA* and *PIK3R2* (Fig. 1a). In contrast, E-cadherin (*CDH1*), an invasion suppressor in many cancers, was upregulated (Table 1 and Fig. 1a). Furthermore, luciferase reporter gene assays showed that ARID4B activated the promoters of *PIK3CA* and *PIK3R2* (Fig. 1b). Consistent with these results, we found a positive correlation between *ARID4B* and *PIK3CA* expression within the prostate cancer cohort in the TCGA database (Fig. 1c).

Furthermore, western blot analysis revealed that knockdown of *ARID4B* reduced protein levels of p110α (encoded by *PIK3CA*) and p85β (encoded by *PIK3R2*) in PC3 cells (Fig. 1d). Accordingly, phosphorylation of the PI3K downstream effector AKT at Ser473, required for the full activation of AKT[29], was reduced in PC3 cells after knockdown of *ARID4B* (Fig. 1d). In addition, phosphorylation of mTOR at Ser2448 was also diminished (Fig. 1d), consistent with reports that phosphorylation of AKT at Ser473 is linked to phosphorylation of mTOR at Ser2448 and activation of the mTOR pathway[30,31]. Concomitantly, expression of downstream target genes in the PI3K/AKT signaling network, such as cyclin D1 (*CCND1*), was decreased (Table 1, Fig. 1a, d Supplementary Fig. 2, and Supplementary Table 2). Consistent with these results, knockdown of *ARID4B* in the DU145 and LNCaP human prostate cancer cell lines reduced expression or phosphorylation of the core regulators and downstream effectors of the PI3K-AKT pathway (Supplementary Fig. 3). Together, these results suggest that ARID4B positively regulates expression

**Table 1 RNA-Seq analysis shows decreased expression of *ARID4B*, *PIK3CA*, *PIK3R2*, and *CCND1*, but increased expression of *CDH1* in PC3 cells with knockdown of *ARID4B* by short interfering RNA (SiARID4B)**

| Gene | log$_2$FC SiARID4B/SiControl | P value |
|---|---|---|
| *ARID4B* | −1.71 | <0.001 |
| *PIK3CA* | −0.69 | 0.004 |
| *PIK3R2* | −0.61 | 0.004 |
| *CCND1* | −0.99 | <0.001 |
| *CDH1* | 1.52 | <0.001 |

Statistical analysis: Wald chi-squared test

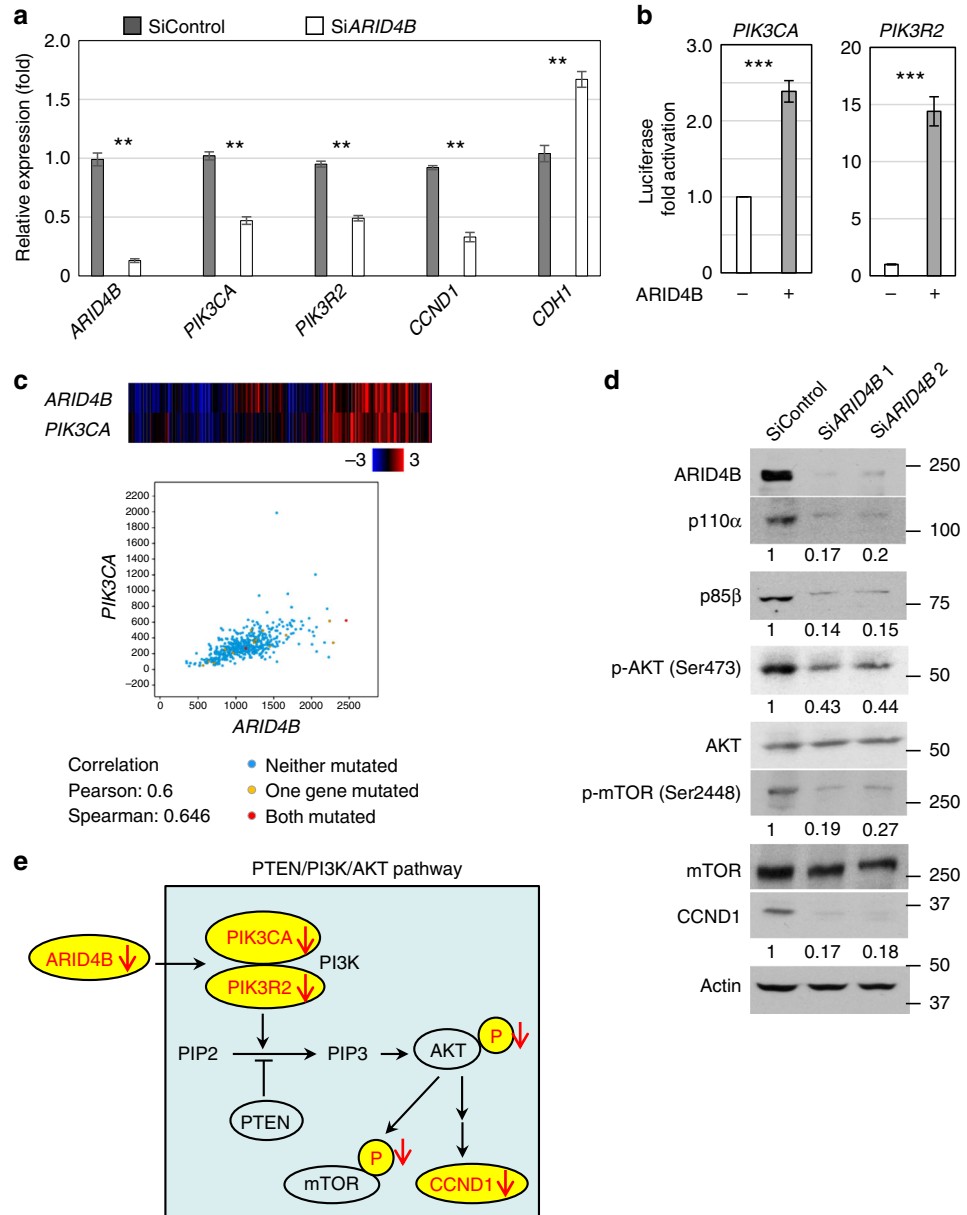

**Fig. 1** Ablation of ARID4B inhibits the PI3K-AKT pathway in prostate cancer cells. **a** mRNA levels of *ARID4B*, *PIK3CA*, *PIK3R2*, *CCND1*, and *CDH1* of PC3 cells transfected with SiControl or SiARID4B were determined by qRT-PCR analyses. The level of gene expression from one sample of PC3 cells transfected with SiControl was set as 1. Data are means ± SEM. **P < 0.01; Statistical analysis: t test. **b** Luciferase reporter assays using the *PIK3CA* (left) or *PIK3R2* (right) promoter-regulated luciferase reporters were performed in HeLa cells with or without overexpression of ARID4B. Data are means ± SEM from three experiments performed in triplicate. ***P < 0.001; Statistical analysis: t test. **c** A regression plot shows the positive correlation between *ARID4B* and *PIK3CA* expression in prostate cancer patients. Data are from the TCGA prostate cancer dataset (n = 492, Provisional). The expression levels of *ARID4B* and *PIK3CA* were presented as mRNA expression z scores. **d** Expression or phosphorylation of core regulators and downstream effectors of the PI3K-AKT pathway in PC3 cells with or without knockdown of *ARID4B* by two different short interfering RNAs SiARID4B 1 or SiARID4B 2 were analyzed by western blots. The intensity of images was measured by ImageJ software. The level of image intensity from the SiControl sample was set as 1. **e** A model of how ARID4B ablation inhibits the PI3K-AKT pathway. Ablation of ARID4B suppresses expression of the PI3K subunits p110α and p85β, which results in dephosphorylation of the PI3K downstream effectors AKT and mTOR and decreased expression of the downstream target CCND1. Downregulated genes and decreased protein phosphorylation are indicated by red arrows

of *PIK3CA* and *PIK3R2* and is crucial for activation of the PI3K/AKT signaling pathway (Fig. 1e).

**PIK3CA and PIK3R2 are direct target genes of ARID4B.** Since RNA-Seq analysis identified genes directly or indirectly regulated by ARID4B (Supplementary Data 1), we performed ChIP using antibody against ARID4B (Supplementary Data 1), we performed ChIP using

antibody against ARID4B followed by DNA sequencing (ChIP-Seq) analysis to globally identify direct binding targets for ARID4B in PC3 cells (Supplementary Data 2). By combining the results from RNA-Seq (P value < 0.05, |Log2FC| ≥ 0.59) and ChIP-Seq analyses, we identified potential genes directly regulated by ARID4B (Supplementary Data 3). ChIP-Seq analysis showed enrichment of ARID4B near the transcriptional start sites

(TSSs) of genes (Supplementary Fig. 4a, b), and identified *PIK3CA* and *PIK3R2*, whose promoters were associated with ARID4B (Fig. 2a, b, orange lines). The CRISPR-Cas9 gene editing system was used to achieve knockout of *ARID4B* (*ARID4B*KO) in PC3 cells (Supplementary Fig. 6a). ChIP-qPCR analysis using the antibody against ARID4B further confirmed recruitment of ARID4B to the *PIK3CA* and *PIK3R2* promoters in control PC3 cells, but showed minimal ARID4B binding in *ARID4B*KO PC3 cells (Fig. 2c). In contrast, ARID4B was not recruited to the promoters of myogenic differentiation 1 (*MYOD1*), glyceraldehyde 3-phosphate dehydrogenase (*GAPDH*), or *ARID4B* itself (Fig. 2c, Supplementary Fig. 4c, and Supplementary Data 2), indicating that recruitment of ARID4B to the *PIK3CA* and *PIK3R2* promoters is specific. Together with the fact that ARID4B regulates promoter activities of *PIK3CA* and *PIK3R2* (Fig. 1b) and controls their expression (Table 1 and Fig. 1a), these results indicate that *PIK3CA* and *PIK3R2* are direct target genes of ARID4B.

**ARID4B and histone H1 reciprocally bind to *PIK3CA* and *PIK3R2*.** Histone H1 is depleted in the region surrounding the TSSs[32], and we showed that ARID4B is recruited to TSSs of genes (Supplementary Fig. 4a, b). Thus, we examined whether recruitment of ARID4B to the *PIK3CA* and *PIK3R2* promoters affects binding of histone H1. ChIP-qPCR analyses showed that ablation of ARID4B increased binding of histone H1 to the *PIK3CA* and *PIK3R2* promoters (Fig. 2d) without affecting histone H4 acetylation (H4Ac), H3 lysine 27 acetylation (H3K27Ac), or H3 lysine 4 trimethylation (H3K4me3) (Supplementary Fig. 5). Consistent with no recruitment of ARID4B to the *MYOD1*, *GAPDH*, and *ARID4B* promoters (Fig. 2c and Supplementary Fig. 4c), ablation of ARID4B did not influence histone H1 binding to these promoters (Fig. 2d and Supplementary Fig. 4d).

**ARID4B modulates chromatin condensation on *PIK3CA* and *PIK3R2*.** Since histone H1 stabilizes the condensed chromatin structure that contributes to gene silencing[33], reciprocal binding of ARID4B and histone H1 to the *PIK3CA* and *PIK3R2* promoters led us to examine whether ARID4B impacts chromatin structure on these promoters. Using micrococcal nuclease (MNase) that preferentially digests loosened chromatin followed by qPCR analysis, the promoter regions of *PIK3CA* and *PIK3R2* (but not *MYOD1*) in control PC3 cells were more sensitive to MNase digestion than *ARID4B*KO PC3 cells (Fig. 2e), indicating that the chromatin was less condensed in the presence of ARID4B. Collectively, these results suggest that reciprocal binding of ARID4B and histone H1 to the *PIK3CA* and *PIK3R2* promoters modulates chromatin condensation. Binding of ARID4B excludes histone H1 from the *PIK3CA* and *PIK3R2* promoters, leading to relaxed chromatin architecture and gene activation (Fig. 2f). Upon ARID4B ablation, histone H1 binds to the promoters and condenses the chromatin structure to suppress gene expression (Fig. 2f). Together, these results suggest a mechanism by which ARID4B activates transcription of *PIK3CA* and *PIK3R2*.

**ARID4B is required for PTEN-deficient prostate cancer.** As a tumor suppressor, PTEN counters the actions of PI3K via its lipid phosphatase activity to govern cellular processes, including cell growth, proliferation, and cell migration[7]. PTEN is frequently disrupted in prostate cancer[1,2]. Loss of PTEN function leads to derepression of the PI3K-AKT pathway that dictates tumor susceptibility and favors cancer progression[7]. To determine the functional significance of our findings that ARID4B is a transcriptional activator of the PI3K subunits *PIK3CA* and *PIK3R2*

(Table 1 and Fig. 1a, b), we investigated the role of ARID4B in the development of prostate cancer harboring PTEN deficiency. By analyzing the Cancer Genome Atlas (TCGA) and other prostate cancer genomic datasets[2,20,24,25,34], we identified that *ARID4B* is consistently retained in prostate carcinoma cohorts with deep deletion of *PTEN*, although deletion of *ARID4B* also occasionally occurs in prostate cancer (Fig. 3a).

Because there tends to be mutual exclusivity between *ARID4B* and *PTEN* deletions in prostate cancer genome (Supplementary Table 3), *ARID4B* might be a synthetic-essential gene for prostate tumorigenesis in the context of PTEN deficiency. Therefore, we examined whether inhibition of ARID4B impairs the tumor-promoting actions in prostate cancer cells driven by loss of PTEN. In PC3 cells, knockout of ARID4B not only efficiently constrained cell growth (Fig. 3b), but also significantly suppressed aggressiveness of cancer cells, including cell migration, cell invasion, and tumorsphere formation (Fig. 3c). In addition, ablation of ARID4B sufficiently inhibited cell proliferation and tumor formation in tumor xenograft experiments (Fig. 3d, e). On the other hand, knockout of *ARID4B* in the PTEN-intact prostate cancer cell line DU145 only moderately suppressed tumor growth and cell proliferation, and not to the same extent as in PC3 cells (Supplementary Fig. 6). Together, these results indicate that *ARID4B* is required for tumor growth of PTEN-null prostate cancer cells.

**Knockout of *Arid4b* compromises prostate cancer progression.** To demonstrate the effects of ARID4B ablation on prostate tumorigenesis driven by PTEN deficiency in mice, we used the well-characterized prostate cancer mouse model that carries prostate-specific deletion of *Pten* (*Pten*$^{PC-/-}$), and generated mice carrying prostate-specific deletions of both *Pten* and *Arid4b* (*Pten*$^{PC-/-}$*Arid4b*$^{PC-/-}$) (Supplementary Fig. 7a). While deletion of *Pten* resulted in enlarged prostates in the *Pten*$^{PC-/-}$ mice, this effect was reduced in the *Pten*$^{PC-/-}$*Arid4b*$^{PC-/-}$ mice (Fig. 4a and Supplementary Fig. 7b). The *Pten*$^{PC-/-}$ mice develop low-grade prostatic intraepithelial neoplasia (PIN) that may progress to high-grade PIN or adenocarcinoma[35], which faithfully mimics prostate cancer progression in humans. Consistent with the earlier report, we showed that most *Pten*$^{PC-/-}$ mice developed high-grade PIN or adenocarcinoma at 5 months of age (Fig. 4b, c and Supplementary Fig. 7c). In contrast, tumor development in prostates of most of the 5-month-old *Pten*$^{PC-/-}$*Arid4b*$^{PC-/-}$ mice was arrested at the normal, hyperplastic, or low-grade PIN stages (Fig. 4b, c and Supplementary Fig. 7c), suggesting that ablation of ARID4B attenuated prostate cancer progression. Correspondingly, there were fewer Ki67-labeled nuclei (a marker of proliferation) in prostates of the *Pten*$^{PC-/-}$*Arid4b*$^{PC-/-}$ mice (Fig. 4d), suggesting that ablation of ARID4B compromised the proliferation advantage elicited by loss of PTEN. These effects were not due to defects in prostate development since the *Arid4b*$^{PC-/-}$ mice had prostates of normal size and histology (Supplementary Fig. 8a, b).

Consistent with the results from PC3 cells with ablation of ARID4B (Fig. 1d), expression or phosphorylation of the core regulators and downstream effectors of the PI3K-AKT pathway were reduced in prostates of *Pten*$^{PC-/-}$*Arid4b*$^{PC-/-}$ mice (Fig. 4e), suggesting a mechanism by which ARID4B is required for executing actions that promote prostate cancer in the context of PTEN deficiency.

To analyze whether ablation of ARID4B delays the onset of prostate cancer caused by PTEN deficiency, we then examined the *Pten*$^{PC-/-}$*Arid4b*$^{PC-/-}$ mice at 9 months of age. While the *Pten*$^{PC-/-}$ mice exhibited enlarged prostates, prostate size was still comparable in the *Pten*$^{PC-/-}$*Arid4b*$^{PC-/-}$ and control mice

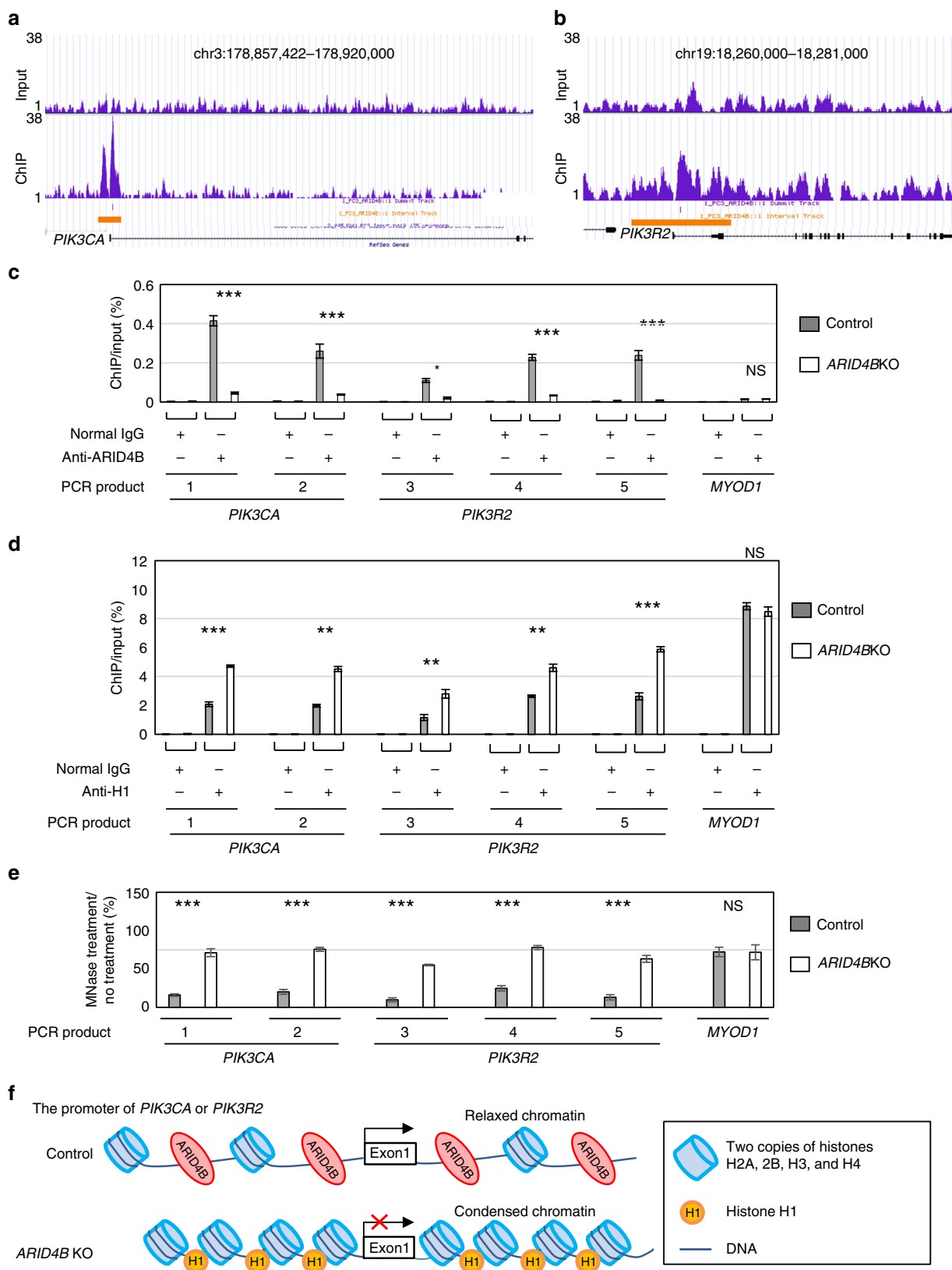

(Supplementary Fig. 9a). Similar to the results from 5-month-old mice, only normal/hyperplasia and PIN were found in the $Pten^{PC-/-}Arid4b^{PC-/-}$ mice (Supplementary Fig. 9b, c). These results suggest that in the $Pten^{PC-/-}Arid4b^{PC-/-}$ mice, PIN did not progress to adenocarcinoma over time. Collectively, these

data demonstrate a synthetic essentiality of ARID4B in prostate cancer elicited by PTEN deficiency.

**PTEN deficiency increases expression of ARID4B.** To better understand the inner workings of the synthetic-essential

**Fig. 2** *PIK3CA* and *PIK3R2* are direct target genes of ARID4B. **a**, **b** ChIP-Seq analysis shows enrichment of ARID4B binding to the promoters of *PIK3CA* (**a**) and *PIK3R2* (**b**) in PC3 cells. Orange lines indicate the ARID4B binding regions. **c**, **d** Recruitment of ARID4B (**c**) and histone H1 (**d**) to the promoters of *PIK3CA*, *PIK3R2*, and *MYOD1* in control and *ARID4B*KO PC3 cells were analyzed by ChIP-qPCR analyses using anti-ARID4B (**c**) or anti-H1 (**d**) antibodies. Normal rabbit IgG was used for comparison. **e** Chromatin accessibility of the *PIK3CA*, *PIK3R2*, and *MYOD1* promoters to micrococcal nuclease (MNase) digestion. The PCR products on the promoter region of each gene detected by specific primer sets in qPCR are as follows: *PIK3CA* (PCR products 1 and 2), *PIK3R2* (PCR products 3–5), and *MYOD1* (**c–e**). See Supplementary Fig. 4e for details on regions of qPCR products. Data are means ± SEM from three experiments performed in triplicate (**c–e**). *$P < 0.05$; **$P < 0.01$; ***$P < 0.001$; NS no significant differences; Statistical analysis: $t$ test (**c–e**). **f** A schematic representation of how ARID4B controls chromatin condensation to regulate the promoter activities of *PIK3CA* and *PIK3R2*. ARID4B maintains a relaxed chromatin structure on the promoters, leading to gene activation. Loss of ARID4B results in occupancy of histone H1 on the promoters and chromatin condensation, leading to suppression of gene transcription

relationship between ARID4B and PTEN, we queried the dataset of transcriptome comparison between wild-type and *Pten*[PC−/−] prostates at 15 weeks of age[36], and found that expression of *Arid4b* was elevated in prostates of the *Pten*[PC−/−] mice (fold change: 4.859; $P = 0.0094$; statistical analysis: $t$ test). We confirmed the increase of *Arid4b* mRNA in the *Pten*[PC−/−] prostates by qRT-PCR analysis (Fig. 5a). Immunohistochemistry and western blot analyses verified a robust increase of ARID4B protein levels in prostates of the *Pten*[PC−/−] mice (Fig. 5b, c). Consistent with these results, expression of *ARID4B* was significantly increased in prostate cancer patients harboring *PTEN* deletion compared to those with unaltered *PTEN* in the TCGA provisional dataset (Fig. 5d). These results suggest that deficiency of PTEN increases expression of ARID4B.

On the other hand, re-expression of *PTEN* in PC3 cells (PTEN-deficient) decreased expression of ARID4B (Fig. 5e), accompanied by lessened abundance of p110α and p85β (encoded by *PIK3CA* and *PIK3R2*, respectively) and inactivation of the PI3K-AKT signaling pathway (Fig. 5e and Supplementary Fig. 10a). Furthermore, *ARID4B* promoter-driven luciferase reporter gene assays showed that knockdown of *PTEN* activated the *ARID4B* promoter in the PTEN-intact prostate cancer cell line DU145 and cervical cancer cell line HeLa (Fig. 5f and Supplementary Fig. 10b), whereas overexpression of PTEN suppressed the *ARID4B* promoter activity (Fig. 5g). Together, these results suggest that PTEN negatively regulates expression of *ARID4B*.

PTEN lipid phosphatase activity dephosphorylates PIP3 into PIP2 to negatively regulate the PI3K signaling cascade. Thus, loss of PTEN promotes PI3K-AKT signaling[6,7]. To demonstrate that PTEN deficiency correlates with increased ARID4B expression via activation of the PI3K-AKT pathway, we showed that knockdown of *AKT* in the PTEN-deficient prostate cell line PC3 decreased expression of ARID4B (Fig. 5h). Consistent with this result, adding an AKT inhibitor MK-2206 downregulated ARID4B in PC3 cells (Fig. 5i). Furthermore, the *ARID4B* promoter-driven luciferase reporter gene assay showed that a constitutively active form of AKT (CA-AKT) activated the promoter of *ARID4B* (Fig. 5j). These results indicate that AKT positively regulates expression of *ARID4B*. Together with the fact that ARID4B is required for activation of the PI3K-AKT signaling pathway (Fig. 1), our results suggest an AKT-ARID4B-positive feedback loop (Fig. 5k).

**Functional interactions between *PTEN*, *ARID4B*, and *PIK3CA*.** To investigate the clinical relevance of the PTEN-ARID4B-PI3K axis in prostate cancer, we performed immunohistochemical staining with antibodies against ARID4B, PTEN, and p110α on tissue microarrays (TMAs) consisting of 118 prostate adenocarcinoma samples from 85 patients and 73 prostate hyperplasia samples from 36 patients. Results revealed that expression of ARID4B was higher in prostate adenocarcinoma compared to prostate hyperplasia (Fig. 6a, left), shown as an increase in the number of ARID4B-positive cells in prostate adenocarcinoma

samples (Fig. 6c, left and Supplementary Fig. 11a, left). This is consistent with results from our Oncomine database analysis, showing increased mRNA levels of *ARID4B* in human prostate carcinoma compared to normal prostate glands (Supplementary Fig. 1b). In prostate adenocarcinoma, PTEN-positive cells were diminished (Fig. 6a, c, middle and Supplementary Fig. 11a, middle), whereas the number of p110α-positive cells was increased (Fig. 6a, c, right and Supplementary Fig. 11a, right). Furthermore, expression of ARID4B and PTEN was negatively correlated in human prostate adenocarcinoma (Fig. 6b, c and Supplementary Fig. 11a), while expression of ARID4B and p110α was positively correlated (Fig. 6b, c and Supplementary Fig. 11a).

To complement our TMA results, we determined how expression of these three genes correlates with prostate cancer recurrence in patients. A multivariate Cox proportional hazards model was used to combine expression levels of *PTEN*, *ARID4B*, and *PIK3CA* with time to tumor recurrence from the dataset GSE40272[28] into an integrated risk score model. We found that the *PTEN*/*ARID4B*/*PIK3CA* three-gene signature had an increased predictive power for prostate cancer recurrence compared to any individual gene alone or two genes combined (Table 2 and Supplementary Table 4). This three-gene signature reliably stratified patients into high- and low-risk groups for tumor recurrence (Fig. 6d). The enhanced prognostic value of the three-gene signature for recurrent prostate cancer was similarly confirmed by analysis of another two independent datasets GSE21032[2] and TCGA provisional (Supplementary Tables 5, 6 and Supplementary Fig. 11b, c). Together, these data support the importance of functional interactions between *PTEN*, *ARID4B*, and *PIK3CA* in human prostate cancer.

## Discussion
Although ARID4B does not have intrinsic methyltransferase or acetyltransferase activities, it contains a Tudor domain and a chromo domain[11]. These two domains may promote assembly of chromatin remodeling complexes[13,14]. ARID4B is a component of the SIN3A/HDAC1 repressor complex that inhibits expression of target genes[12]. In this report, we showed that ARID4B was recruited to the promoters of *PIK3CA* and *PIK3R2* and activated their expression. These results are consistent with our previous findings that ARID4B functions as coactivator (not a co-repressor) for androgen receptor in the testis and is essential for testis development[37].

In addition, while knockdown of *ARID4B* decreased expression of p110α and p85β in the prostate cancer cell lines PC3, DU145, and LNCaP (Fig. 1d and Supplementary Fig. 3), knockdown of *HDAC1* increased expression of p110α and p85β (Supplementary Fig. 12). These results indicate that ARID4B and HDAC1 have opposite effects on regulation of *PIK3CA* and *PIK3R2* expression. As a repressor, HDAC1 deacetylates histones H4 and H3 to inhibit transcriptionally active states of the gene promoters. While HDAC1 suppresses expression of *PIK3CA* and *PIK3R2*, ablation of ARID4B in PC3 cells did not affect levels of H4

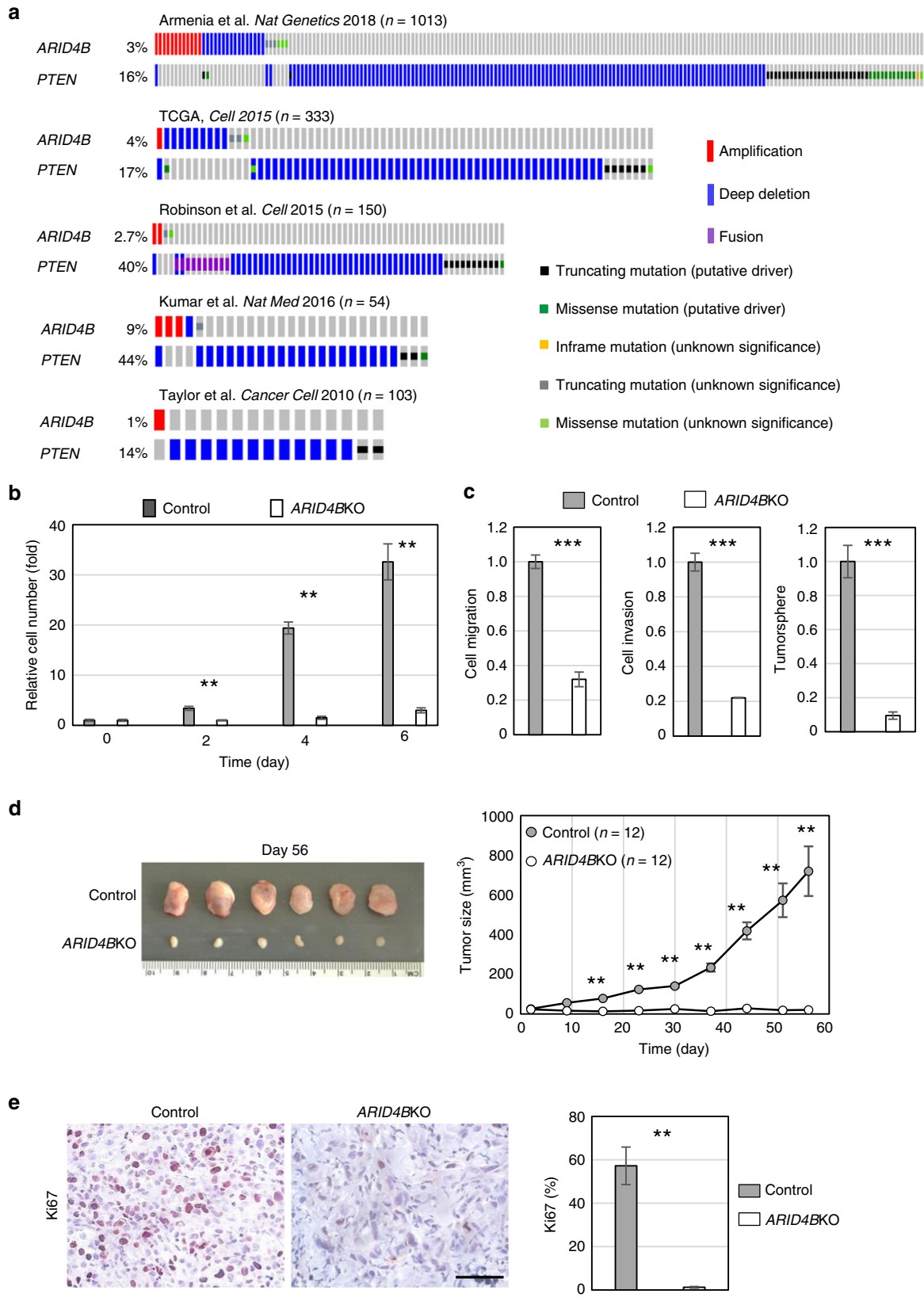

acetylation on the promoters of *PIK3CA* and *PIK3R2* (Supplementary Fig. 5a), suggesting that ARID4B does not mediate HDAC1 activity on these two promoters. Together, our results suggest that ARID4B and HDAC1 are likely to function independently to regulate *PIK3CA* and *PIK3R2* in prostate cancer cells. Therefore, the SIN3A/HDAC1 repressor complex does not contribute to activation of *PIK3CA* and *PIK3R2* by ARID4B. The function of ARID4B in transcriptional regulation may depend on the gene promoter context and the proteins that associate with ARID4B, which could dictate its function as an activator

**Fig. 3** ARID4B is required for tumor growth of PTEN-deficient prostate cancer. **a** Genomic alterations of *ARID4B* and *PTEN* in prostate adenocarcinoma from five prostate cancer genomic datasets. **b** Proliferation of control and *ARID4B*KO PC3 cells was measured by MTT assays. **c** Comparisons of cell migration, cell invasion, and tumorsphere formation between control and *ARID4B*KO PC3 cells. **d** Representative images of excised tumors (left) and measurement of tumor growth (right) from xenografts in mice injected with control or *ARID4B*KO PC3 cells. Data are means ± SEM (right). **e** Representative images (left) and quantification (right) of immunohistochemical staining with Ki67 (cellular marker of proliferation) on excised tumors from mice injected with control or *ARID4B*KO PC3 cells (*n* = 3 for each group). Scale bar, 50 mm. Data are means ± SEM from three experiments performed in triplicate (**b**, **c**, **e**). \*\**P* < 0.01 (**b**, **d**, **e**); \*\*\**P* < 0.001 (**c**); Statistical analysis: *t* test

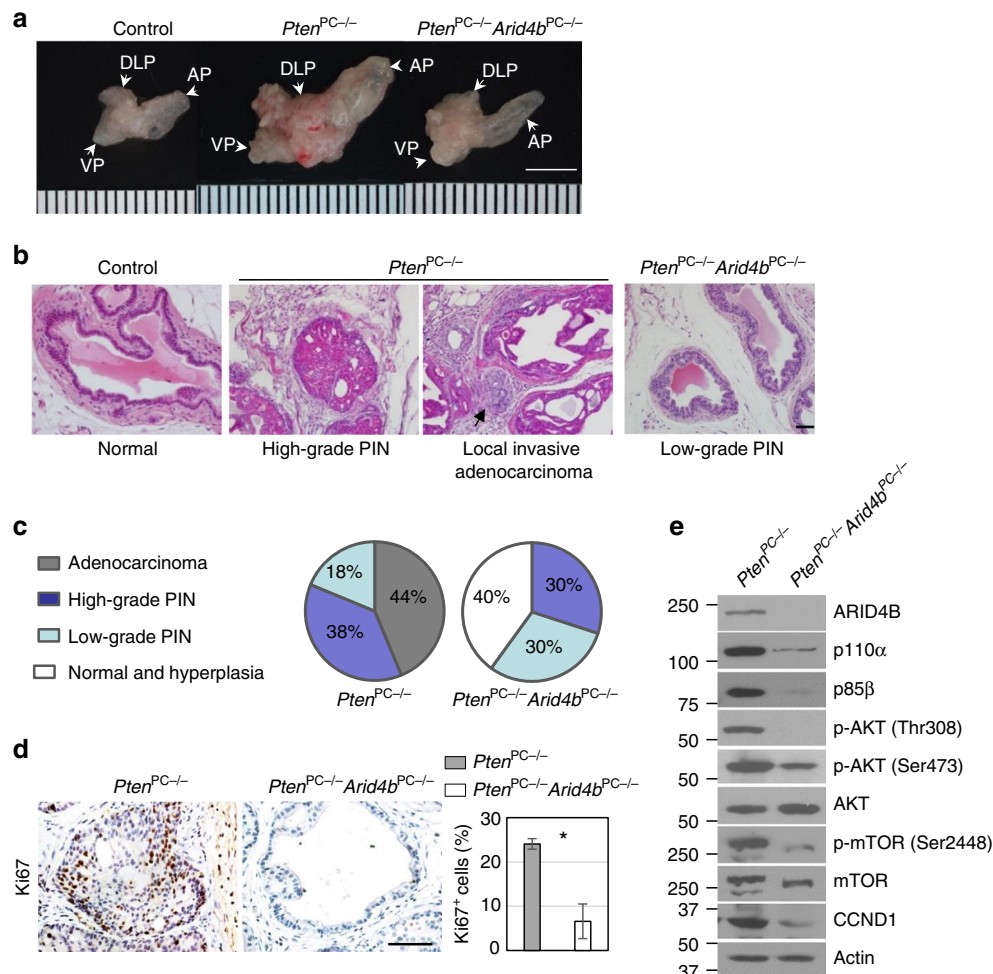

**Fig. 4** Knockout of *Arid4b* in mice compromises cancer progression in PTEN-deficient prostates. **a** Gross anatomy of prostates from the control, *Pten*$^{PC-/-}$, and *Pten*$^{PC-/-}$*Arid4b*$^{PC-/-}$ mice at 5 months of age. AP anterior prostate, VP ventral prostate, DLP dorsal-lateral prostate. Scale bar, 5 mm. **b** Haematoxylin and eosin staining of the dorsal prostates from the control, *Pten*$^{PC-/-}$, and *Pten*$^{PC-/-}$*Arid4b*$^{PC-/-}$ mice at 5 months of age. Arrow: local invasive adenocarcinoma in the *Pten*$^{PC-/-}$ prostate. PIN prostatic intraepithelial neoplasia. Scale bar, 50 μm. **c** Pie graphs are used to summarize prostate tumor progression in the *Pten*$^{PC-/-}$ (*n* = 16) and *Pten*$^{PC-/-}$*Arid4b*$^{PC-/-}$ mice (*n* = 10) at 5 months of age. **d** Representative images (left) and quantification (right) of immunohistochemical Ki67 staining cells in prostates from the *Pten*$^{PC-/-}$ and *Pten*$^{PC-/-}$*Arid4b*$^{PC-/-}$ mice at 5 months of age. Three mice from each genotype were analyzed. Data are means ± SEM. \**P* < 0.05; Statistical analysis: *t* test. Scale bar, 50 μm. **e** Expression or phosphorylation of core regulators and downstream effectors of the PI3K-AKT pathway in anterior prostate of the *Pten*$^{PC-/-}$ and *Pten*$^{PC-/-}$*Arid4b*$^{PC-/-}$ mice at 7 weeks of age were analyzed by western blots

or a repressor. Identifying ARID4B-associated proteins could provide mechanistic insight into how ARID4B regulates gene transcription.

Analysis of five prostate cancer genome datasets using the cBioportal data source (http://www.cbioportal.org/) suggest that *ARID4B* and *PTEN* deletions tend to be mutually exclusive in the prostate cancer genome, although the relationship did not reach statistical significance (*P* > 0.05) (Supplementary Table 3). In most cohorts, the *ARID4B* gene alteration rate is less than 5%,

which makes it difficult to accrue enough samples for a significant *P* value. Similar to our findings, others have reported a tendency of mutual exclusivity between *CHD1* and *PTEN* deletions in prostate cancer without reaching statistical significance[5]. In addition, although the mutual exclusivity between *ARID1B* and *KRAS* is not statistically significant in any individual type of cancer, statistical significance for mutual exclusivity was achieved when multiple cancer types were combined to increase the sample size[38]. The tendency of ARID4B-PTEN to be mutually exclusive

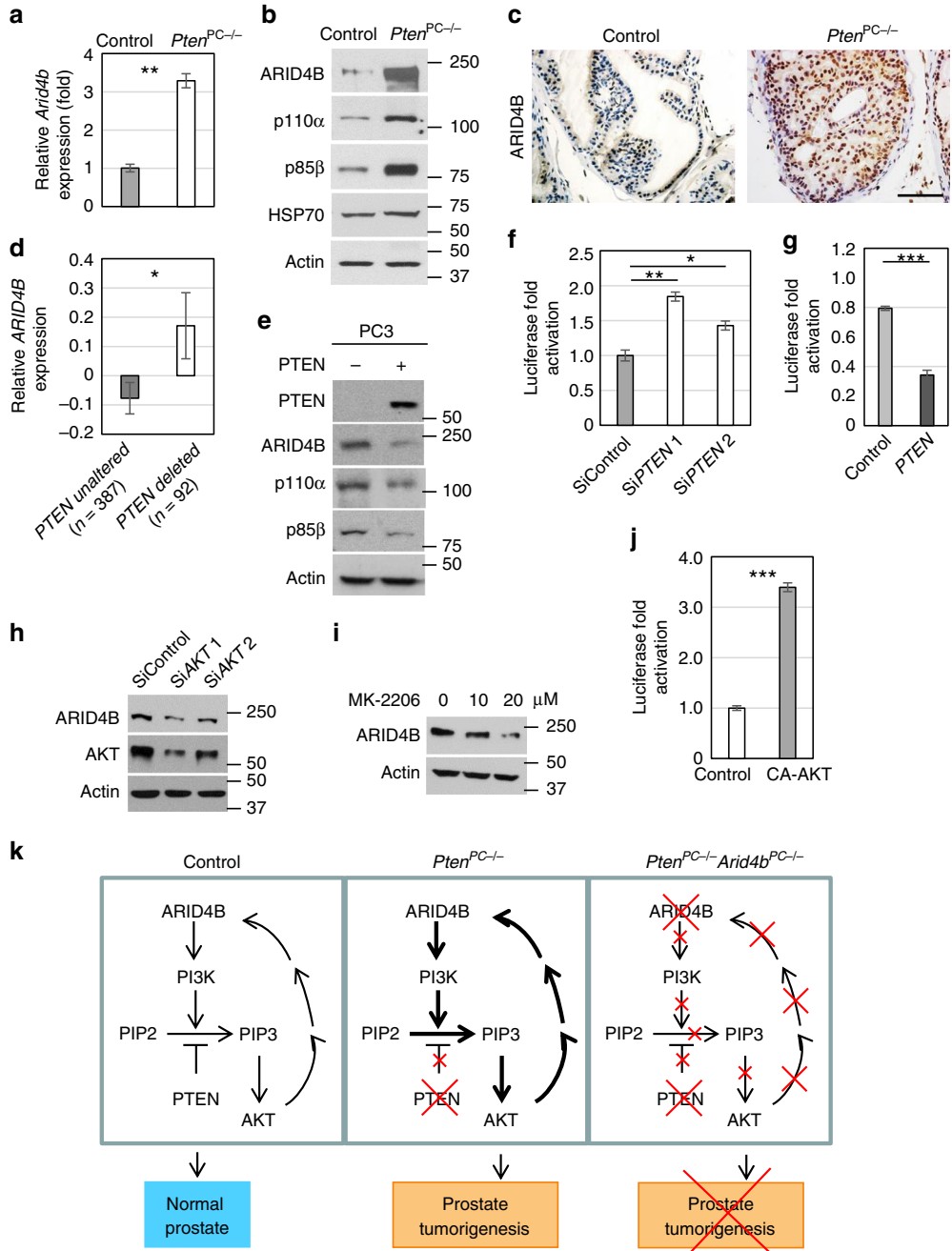

**Fig. 5** PTEN deficiency increases expression of ARID4B via an AKT-ARID4B feedback loop. **a** Levels of *Arid4b* mRNA in anterior prostate from 3-month-old control and *Pten*^PC−/− mice were measured by qRT-PCR analysis. **b**, **c** Levels of ARID4B, p110α, p85β, HSP70, and actin in control and *Pten*^PC−/− anterior prostate at 3 months of age were determined by western blot analysis (**b**) or immunohistochemical staining (**c**). Scale bar, 50 μm. **d** Relative *ARID4B* expression levels in human prostate cancers with unaltered or deleted *PTEN* from the TCGA dataset (provisional). Data are means ± SEM. **e** Levels of PTEN, ARID4B, p110α, p85β, and actin in PC3 cells with or without PTEN re-expression were determined by western blot analysis. **f**, **g** Luciferase reporter assays using the *ARID4B* promoter-driven luciferase reporter were performed in DU145 cells with or without knockdown of *PTEN* by two different short interfering RNAs Si*PTEN* 1 or Si*PTEN* 2 (**f**), and in HeLa cells with or without PTEN overexpression (**g**). **h**, **i** Western blot analyses determined the levels of ARID4B and AKT in PC3 cells with or without knockdown of *AKT* by short interfering RNAs Si*AKT* 1 or Si*AKT* 2 (**h**), or in PC3 cells untreated or treated with an AKT inhibitor MK-2206 (**i**). **j** Luciferase reporter assay using the *ARID4B* promoter-driven luciferase reporter was performed in HeLa cells with or without overexpression of CA-AKT (constitutively active AKT). Data are means ± SEM from three experiments performed in triplicate (**a**, **f**, **g**, **j**). *$P < 0.05$; **$P < 0.01$; ***$P < 0.001$; Statistical analysis: *t* test (**a**, **d**, **f**, **g**, **j**). **k** Genetically engineered mouse models demonstrate that ARID4B is a key factor in the PTEN/ PI3K pathway to promote tumorigenesis in prostate with PTEN deficiency. In normal prostate (left), PTEN opposes the PI3K/AKT signaling pathway and constrains prostate tumorigenesis. In *Pten*^PC−/− mice (middle), PTEN ablation results in activation of the PI3K/AKT pathway and the AKT-ARID4B positive feedback loop, leading to prostate tumorigenesis. In *Pten*^PC−/−*Arid4b*^PC−/− mice (right), knockout of *ARID4B* suppresses PI3K expression to compromise activation of the PI3K/AKT pathway, thus preventing prostate tumorigenesis elicited by PTEN deficiency

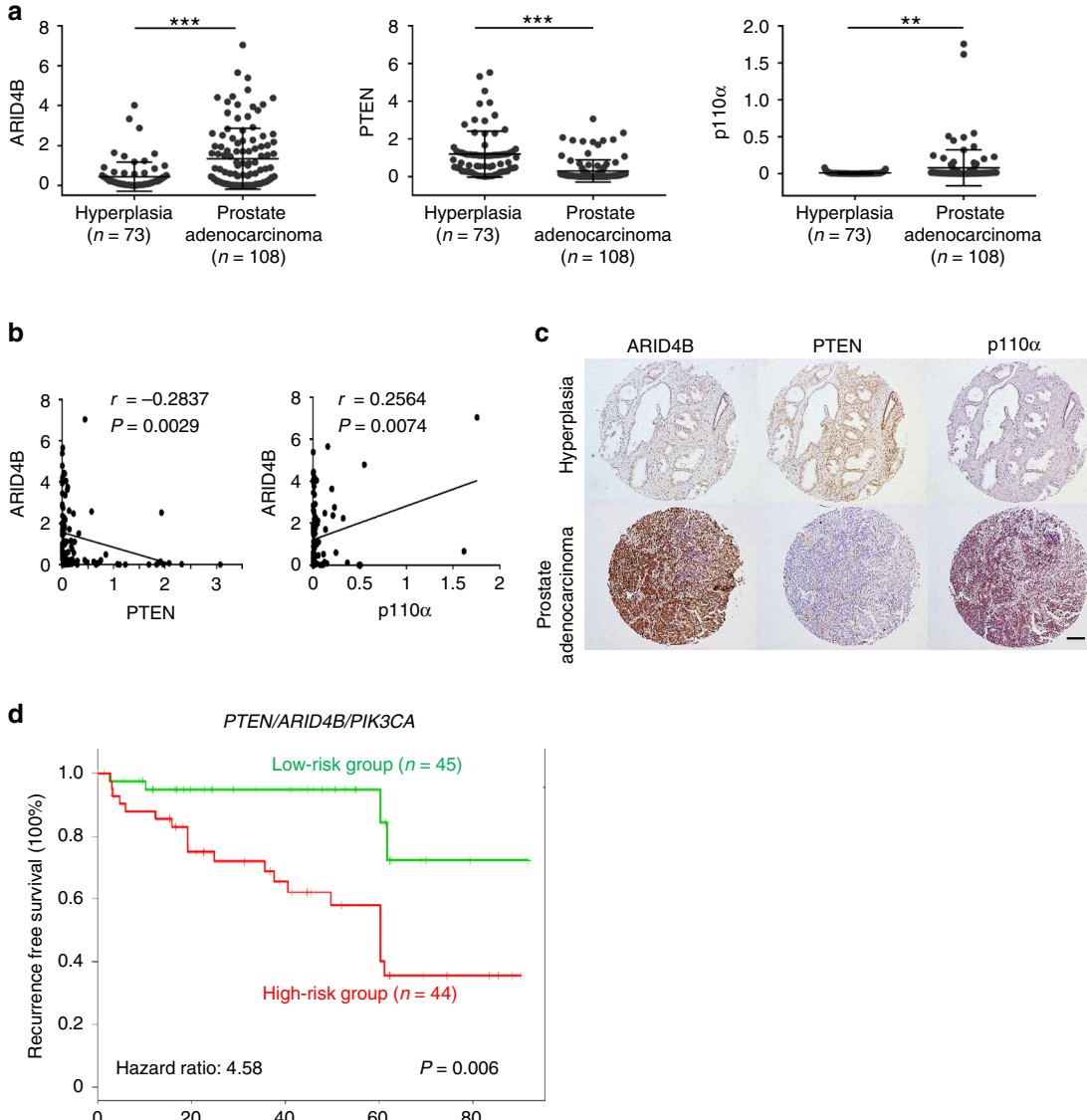

**Fig. 6** Prognostic potential of the *PTEN/ARID4B/PIK3CA* signature in human prostate cancer. **a** Expression of ARID4B, PTEN, and p110α between human prostate hyperplasia and prostate adenocarcinoma in tumor microarrays was compared. Seventy-three prostate hyperplasia samples from 36 patients and 118 prostate adenocarcinoma samples from 85 patients were analyzed. The expression levels of ARID4B, PTEN, and p110α were presented as the numbers of positive cells (×1000) per mm$^2$. **b** Regression plots show expression correlations of ARID4B with PTEN and p110α in prostate adenocarcinoma from tumor microarray results. In all, 118 prostate adenocarcinoma samples from 85 patients were analyzed. The expression levels of ARID4B, PTEN, and p110α were presented as the numbers of positive cells (×1,000) per mm$^2$. **c** Representative immunohistochemical staining of ARID4B, PTEN, and p110α from one patient with prostate hyperplasia (top) and one with prostate adenocarcinoma (bottom) in tumor microarrays. Scale bar, 200 μm. **d** Kaplan−Meier plot of prostate cancer recurrence-free survival in patients stratified by the *PTEN/ARID4B/PIK3CA* signature. Data are from the GSE40272 dataset. Statistical analysis: log-rank test

led us to further evaluate the relationship between these two genes. Extensive in vitro and in vivo data, including those from depletion of ARID4B in the PTEN-deficient prostate cancer cell line PC3 and the prostate-specific *Arid4b* and *Pten* double knockout mouse model, demonstrated that PTEN-deficient prostate cancer depends on ARID4B.

In both PC3 and DU145 cells, depletion of *ARID4B* resulted in reduced expression or phosphorylation of the core regulators and downstream effectors of the PI3K-AKT pathway, while expression of PTEN in DU145 cells was unchanged (Supplementary Fig. 3a). We showed that depletion of ARID4B inhibited colony formation in PC3 and DU145 cells (Supplementary Fig. 13a, b), and re-expression of ARID4B in the *ARID4B*KO PC3 and

*ARID4B*KO DU145 cells increased colony formation (Supplementary Fig. 13c). Depletion of *ARID4B* in another PTEN-deficient prostate cancer cell line, LNCaP, significantly suppressed cell growth (Supplementary Fig. 14). In the PC3 (PTEN-deficient) cells, knockout of *ARID4B* efficiently inhibited cancer cell growth and tumor formation in xenografts, suggesting that PTEN-deficient prostate cancer cells depend on ARID4B. In the DU145 (PTEN-intact) cells, knockout of *ARID4B* also suppressed cell growth and tumor formation in xenografts, but less than in PC3 cells. PTEN-deficient PC3 cells may depend more on activation of the PI3K-AKT pathway for tumor growth than the PTEN-intact DU145 cells. This would explain why the AKT inhibitor MK-2206 inhibited cell growth more effectively in PC3 cells than

**Table 2 PTEN plus ARID4B and PIK3CA increased predictive power for prostate cancer recurrence in multivariate Cox regression analysis using the human prostate cancer dataset GSE40272 ($n = 89$)**

| Multivariate Cox | P value | HR | 95% CI |
|---|---|---|---|
| *ARID4B* | 0.059 | 2.34 | 0.97–5.64 |
| *PIK3CA* | 0.790 | 0.9 | 0.4–2.01 |
| *PTEN* | 0.379 | 1.44 | 0.64–3.28 |
| *ARID4B* + *PIK3CA* | 0.05 | 2.41 | 1–5.82 |
| *PTEN* + *ARID4B* | 0.010 | 4.11 | 1.4–12.06 |
| *PTEN* + *PIK3CA* | 0.72 | 1.16 | 0.51–2.62 |
| *PTEN* + *ARID4B* + *PIK3CA* | 0.006 | 4.58 | 1.56–13.42 |

Statistical analysis: log-rank test

DU145 cells (Supplementary Fig. 15a, b). In PTEN-deficient PC3 cells, ablation of PTEN activates the PI3K/AKT pathway, leading to tumorigenesis (Supplementary Fig. 15c, left). In *ARID4B*KO PC3 cells, knockout of *ARID4B* compromises activation of the PI3K/AKT pathway, thus efficiently inhibiting tumorigenesis elicited by PTEN deficiency (Supplementary Fig. 15c, right). In PTEN-intact DU145 cells, PTEN dephosphorylates PIP3 into PIP2, which opposes the PI3K/AKT signaling pathway (Supplementary Fig. 15d, left), suggesting tumorigenesis of DU145 cells is less dependent on the PI3K-AKT signaling than PC3 cells. Therefore, knockout of *ARID4B* that inactivates PI3K/AKT pathway only moderately suppressed tumorigenesis of DU145 cells (Supplementary Fig. 15d, right).

In addition, ARID4B may promote cancer cell activity through additional pathways unrelated to PTEN tumor suppressor function. For example, our immunohistochemical analyses detected increased numbers of cells positive for cleaved caspase 3, an apoptosis marker, in prostates of the *Pten*PC−/−*Arid4b*PC−/− mice compared to the *Pten*PC−/− mice, suggesting that ablation of ARID4B promoted apoptosis (Supplementary Fig. 16a, b). Consistent with this result, expression levels of the apoptosis regulators *p53* and *Fas* were elevated in prostates of the *Pten*PC−/−*Arid4b*PC−/− mice compared to the *Pten*PC−/− mice (Supplementary Fig. 16c). For its oncogenic function, ARID4B might directly or indirectly inhibit the apoptosis pathway mediated by the p53 tumor suppressor.

Similar to the *PTEN/ARID4B/PIK3CA* three-gene signature, *PTEN* together with *SMAD*, *CCND1*, and *SPP1* form a four-gene signature that predicts recurrence and lethal metastasis in human prostate cancer[36]. SMAD4 is a key regulator of the TGFβ pathway and a suppressor of prostate tumor progression, whereas cyclin D1 and SPP1 act as mediators of the prostate cancer processes. Inactivation of PTEN and SMAD4 and activation of cyclin D1 and SPP1 promote prostate cancer. The combination of PTEN/ SMAD4 does not predict risk for prostate cancer recurrence, but the *PTEN/SMAD4/CCND1/SPP1* four-gene signature does. This four-gene model distinguished patient groups at high versus low risk for prostate cancer recurrence by Kaplan–Meier analysis [hazard ratio (HR) = 2.6, $P = 0.012$][36]. In addition, the PTEN/SMAD4/CCND1/SPP1 four-gene signature involves the PTEN and TGFβ pathways, which converges to regulate prostate cancer progression[36]. Similarly, we showed that the *PTEN/ARID4B/PIK3CA* signature has greater predictive power for tumor recurrence than any individual gene alone or two genes combined. This *PTEN/ARID4B/PIK3CA* signature robustly stratified patients into those at high versus low risk for tumor recurrence (HR = 4.58, $P = 0.006$ in the GSE40272 dataset; HR = 3.23, $P = 0.002$ in the GSE21032 dataset; HR = 1.51, $P = 0.025$ in the TCGA dataset; Statistical analysis: log-rank test). Thus, we identified that the PTEN-ARID4B-PI3K regulatory axis underlies the predictive power of the *PTEN/ARID4B/PIK3CA* signature for tumor recurrence in prostate cancer patients.

In this study, our data suggest that ARID4B, by directly impinging on the PTEN-PI3K signaling pathway, functions as a synthetic-essential effector for prostate cancer initiation and progression elicited by loss of PTEN. Our study reveals the PTEN-ARID4B-PI3K axis and an AKT-ARID4B-positive feedback loop whereby PTEN antagonizes the PI3K-AKT signaling pathway. In normal prostate, PTEN is attributed to its lipid phosphatase activity that directly counters PI3K actions, whereas ARID4B is a transcriptional activator of the PI3K subunit genes *PIK3CA* and *PIK3R2*, and expression of *ARID4B* itself is regulated by the AKT-ARID4B-positive feedback loop (Fig. 5k, left). Deficiency of PTEN activates the PI3K-AKT pathway and the AKT-ARID4B-positive feedback loop, leading to prostate tumorigenesis (Fig. 5k, middle). In contrast, ablation of ARID4B inhibits the PI3K-AKT pathway, and compromises cancer progression in prostate harboring PTEN deficiency (Fig. 5k, right). Therefore, our study demonstrates the synthetic-essential role of ARID4B in PTEN-deficient prostate cancer, and provides the underlying mechanism by which ARID4B is required for executing prostate cancer-promoting actions in the context of PTEN deficiency. Together, our findings reveal insights into how PTEN controls prostate tumorigenesis, and support the concept that ARID4B is a potential target for interventions in patients with prostate cancer carrying *PTEN* mutations.

## Methods
**Mice**. Mice with prostate-specific deletion of *Pten* (*Pten*PC−/−) contain the *Pb*-Cre (the *Cre* gene under control of the prostate-specific probasin promoter)[39] and conditional knockout *Pten*flox/flox[35] alleles. The *Pten*flox/flox mice[35] were obtained from Dr. Hong Wu at the University of California Los Angeles. The *Pb*-Cre transgenic mice[39] were obtained from Dr. Fen Wang at the Institute of Bioscience and Technology, Texas A&M Health Science Center. The *Pten*flox/flox mice and *Pb*-Cre mice were backcrossed to generate the *Pten*PC−/− mice. The *Pten*PC−/− mice were further backcrossed with the *Arid4b*flox/flox mice to generate the *Pten*PC−/−*Arid4b*PC−/− mice that carry prostate-specific deletions of *Pten* and *Arid4b* containing the *Pb*-Cre[39], *Pten*flox/flox[35], and *Arid4b*flox/flox[37] alleles. The *Arid4b*flox/flox mice were previously generated by our laboratory[37]. We have complied with all relevant ethical regulations for animal research. Protocols were approved by the Institutional Animal Care and Use Committee (IACUC) of the George Washington University (protocol number: A315). All mice were bred and maintained at the institution's specific pathogen-free mouse facility. The facility is approved by the American Association for Accreditation of Laboratory Animal Care and operated in accordance with current regulations and standards of the US Department of Agriculture and the Department of Health and Human Services.

**Histology and immunohistochemistry**. The prostates were dissected from mice and fixed in 4% paraformaldehyde in phosphate-buffered saline (PBS) for 24 h, processed, and embedded in paraffin. Histological analysis was performed on 5-μm-thick paraffin-embedded prostate sections by hematoxylin and eosin staining. For immunohistochemistry, antigen retrieval was performed by boiling the paraffin-embedded prostate sections in citric acid-based antigen unmasking solution (Vector Laboratories). Samples were blocked with blocking solution (5% donkey serum, 2% bovine serum albumin, and 0.02% Triton-X 100 in PBS), followed by incubation with primary antibodies in blocking solution. The primary antibodies used are anti-ARID4B (A302-233A, Bethyl Laboratories; dilution 1: 1000), anti-Ki67 (B56; 550609, BD Biosciences; dilution 1: 50), and anti-cleaved caspase 3 (9661, Cell Signaling; dilution 1: 400) antibodies. Sections were washed with 0.1% Triton X-100/PBS buffer, and incubated with biotinylated secondary antibodies (Jackson ImmunoResearch). Signal detection was carried out using the ImmPACT DAB Peroxidase Substrate system (Vector Laboratories). Sections were counterstained with hematoxylin (Sigma). Images were captured with a Nikon Eclipse T*i*-U microscope.

**Immunohistochemical staining of tissue microarrays (TMAs)**. Immunohistochemical staining of two TMAs (PRC1021 and PRD961) and image scanning services were provided by Pantomics (Pantomics, Inc. USA). Two TMA slides, PRC1021 and PRD961, included 118 prostate adenocarcinoma cores from 85 patients, 73 prostate hyperplasia cores from 36 patients, and 5 benign tumor cores. These TMA slides were baked at 60 °C for 30 min and then deparaffinized. After antigen retrieval, slides were incubated with primary antibodies: anti-ARID4B (A302-233A, Bethyl Laboratories; dilution 1:400), anti-PTEN (A2B1; sc-7974,

Santa Cruz Biotechnology; dilution 1:100), and anti-p110α (C73F8; 4249, Cell Signaling Technology; dilution 1:200). Slides were washed with 0.1% Triton X-100/PBS buffer, and incubated with the secondary antibody conjugated with horseradish peroxidase. Signal detection was carried out using DAB (3,3′-diaminobenzidine) substrate. Slides were counterstained with hematoxylin. The scanned images were analyzed with QuPath (v0.1.2)[40]. Expression of ARID4B, PTEN, and p110α is presented as the number of positive epithelial cells per mm$^2$.

**Western blot analysis.** PC3 and DU145 cells were lysed in lysis buffer [20 mM Tris·HCl (pH 8.0), 150 mM NaCl, 0.5% Nonidet P-40, 2 mM EDTA, protease inhibitor mixture (cOmplete™ Protease Inhibitor Cocktail, Roche), and phosphatase inhibitor mixture (PhosSTOP™, Roche)] for 30 min, followed by centrifugation at 13,400 × $g$ for 20 min at 4 °C to clear debris. Prostates were lysed in RIPA buffer [25 mM Tris·HCl (pH 7.6), 150 mM NaCl, 1% Nonidet P-40, 1% sodium deoxycholate, 0.1% SDS, protease inhibitor mixture (cOmplete™ Protease Inhibitor Cocktail, Roche), and phosphatase inhibitor mixture (PhosSTOP™, Roche)]. For western blot analysis, the samples were resolved by SDS/PAGE and transferred to nitrocellulose membranes (Bio-Rad). After blocking with 5% milk in TBST buffer [20 mM Tris·HCl, 150 mM NaCl (pH 7.5), 0.1% Tween 20], the primary antibodies diluted in TBST buffer with 5% milk were added to the membranes overnight at 4 °C, followed by incubation with the appropriate horseradish peroxidase-conjugated secondary antibodies (Jackson ImmunoResearch; dilution 1: 5000) for 1 h at room temperature. Signal detection was carried out by Supersignal substrate (Supersignal West Pico Plus Luminol/Enhancer Solution, Thermo Scientific). Subsequent probing with different antibodies was made possible by stripping the membranes with buffer [62.5 mM Tris·HCl (pH 6.8), 2% SDS, 100 mM β-mercaptoethanol] at 55 °C for 30 min. The primary antibodies used for western blotting were: anti-ARID4B (A302-233A, Bethyl Laboratories; dilution 1:2000), anti-p110α (C73F8; 4249, Cell Signaling Technology; dilution 1:1000), anti-p85β (T15; 56934, Santa Cruz Biotechnology; dilution 1:1000), anti-phospho-AKT (Thr308) (9275, Cell Signaling Technology; dilution 1:1000), anti-phospho-AKT (Ser473) (9271, Cell Signaling Technology; dilution 1:1000), anti-AKT (9272, Cell Signaling Technology; dilution 1:1000), anti-phospho-mTOR (Ser2448) (2971, Cell Signaling Technology; dilution 1:1000), anti-mTOR (7C10; 2983, Cell Signaling Technology; dilution 1:1000), anti-CCND1 (M-20; sc-718, Santa Cruz Biotechnology; dilution 1:500), anti-AR (N-20; sc-816, Santa Cruz Biotechnology; dilution 1:500), anti-HSP70 (3A3; sc-32239, Santa Cruz Biotechnology; dilution 1:1000), anti-HDAC1 (05-614, Upstate; dilution 1:1000), anti-PTEN (A2B1; sc-7974, Santa Cruz Biotechnology; dilution 1:500), and anti-actin (AC-74; A2228; Sigma; dilution 1:2000). The blots were cropped with at least one marker present to indicate the molecular weight position. The uncropped scans of all blots are shown in the Source Data file.

**RNA-Seq and qRT-PCR analyses.** Total RNA was purified from control (SiControl) and ARID4B-knockdown (SiARID4B) PC3 cells using an RNeasy plus kit (Qiagen), and was treated by DNase I (Qiagen). Purified RNA was processed for RNA-Seq and initial data analysis (Q$^2$ Solutions). For qRT-PCR analysis, 2 µg of DNase I-treated total RNA was used for reverse transcription to synthesize the first-strand cDNA by the Superscript IV First-strand synthesis system (Invitrogen). qPCR was performed on the ABI StepOne Plus machine using TaqMan gene expression assays (Applied Biosystems). The TaqMan primer/probe sets for genes (Applied Biosystems) are as follows: *ARID4B* (Hs00249610_m1), *PIK3CA* (Hs00907957_m1), *PIK3R2* (Hs00178181_m1), *CCND1* (Hs00765553_m1), *CDH1* (Hs01023894_m1), *ADD3* (Hs00249890_m1), *FAM129A* (Hs00223000_m1), *IGFBP5* (Hs00181213_m1), *PSAT1* (Hs00795278_mH), *KDM6B* (Hs00996325_g1), *FOSB* (Hs00171851_m1), *GAPDH* (Hs03929097_g1), *Arid4b* (Mm01134036_m1), *Pik3ca* (Mm00435673_m1), *Pik3r2* (Mm00435694_m1), *Ccnd1* (Mm00432359_m1), *p53* (Mm01731290_g1), *Fas* (Mm01204974_m1), and *Gapdh* (Mm99999915_g1). RNA from three samples of each group were analyzed. Levels of gene expression were normalized against expression levels of *GAPDH* or *Gapdh* in each sample. In each experiment, the normalized level of the gene of interest from one of the control samples was set as 1.

**ChIP-Seq analysis.** PC3 cells were cross-linked with 4% formaldehyde at 37 °C for 10 min, and the reaction was quenched by 0.125 M glycine for 5 min. Then, cells were processed for ChIP, library preparation, sequencing, and initial data analysis (Active Motif). ChIP assays were performed using the anti-ARID4B antibody (A302-233A, Bethyl Laboratories; dilution 1:100). Input DNA was used for control. Gene calling was based on presence of the ARID4B binding site with 10 kb of the gene margin.

**ChIP-qPCR analysis.** ChIP assays were performed as previously described[37]. PC3 cells with or without CRISPR-mediated knockout of *ARID4B* (*ARID4B*KO and control, respectively) were used for ChIP-qPCR analysis. Chromatin extracted from cells was immunoprecipitated with normal rabbit IgG (P120-101, Bethyl Laboratories; dilution 1:100), anti-ARID4B (A302-233A, Bethyl Laboratories; dilution 1:100), anti-histone H1 (AE-4, sc-8030, Santa Cruz Biotechnology; dilution 1:50), anti-H4AC antibody (06-866, EMD Millipore; dilution 1:100), anti-H3K27Ac antibody (ab4729, Abcam; dilution 1:100), or anti-H3K4me3 antibody

(07-473, EMD Millipore; dilution 1:100) antibodies. DNA from immunoprecipitated chromatin was analyzed by qPCR using the following primer sets:

*PIK3CA* (PCR product 1)
*PIK3CA*-PF1: 5′-CAGAGGGCTGTGACAGTGCATTC-3′
*PIK3CA*-PR1: 5′-GTGGCTAACCAGCGTGGTTATG-3′
*PIK3CA* (PCR product 2)
*PIK3CA*-PF2: 5′-CTCTCCTAGCTGCAGAGGCGAG-3′
*PIK3CA*-PR2: 5′-CGCACTTCCTCAACCTCCAGC-3′
*PIK3R2* (PCR product 3)
*PIK3R2*-PF1: 5′-CTGTTTGCTGAGCTTGAGAACGC-3′
*PIK3R2*-PR1: 5′-GAAATGGCCGAACACTCTCTACG-3′
*PIK3R2* (PCR product 4)
*PIK3R2*-PF2: 5′-GTCCAAGATACCGGACTGGAGC-3′
*PIK3R2*-PR2: 5′-GCTCACACAGACGCGCACTTG-3′
*PIK3R2* (PCR product 5)
*PIK3R2*-PF3: 5′-GTGACTGAAGGATGCTCAGAGC-3′
*PIK3R2*-PR3: 5′-GGATCATCACAGCGGATAGGAAG-3′
*MYOD1*
*MYOD1*-PF1: 5′-GTTCCTATTGGCCTCGGACT-3′
*MYOD1*-PR1: 5′-CCCGGCTGTAGATAGCAAAG-3′
*GAPDH*
*GAPDH*-PF1: 5′-CGGCTACTAGCGGTTTTACG-3′
*GAPHD*-PR1: 5′-GTCGAACAGGAGGAGCAGAG-3′
*ARID4B*
*ARID4B*-PF1: 5′-GACCAATAGGAACAGTGTATAGGC-3′
*ARID4B*-PR1: 5′-GACTCGACTGACGGGCAAACATC-3′

**Micrococcal nuclease (MNase) sensitivity assay.** Micrococcal nuclease assay was performed as described[41]. Briefly, control and *ARID4B*KO PC3 cells were resuspended in ice-cold lysis buffer [10 mM Tris·HCl (pH 7.4), 10 mM NaCl, 3 mM MgCl$_2$, 0.5% Nonidet P-40, 0.15 mM spermine and 0.5 mM spermidine], and the nuclei were repelleted. Nuclei were washed with MNase digestion buffer [10 mM Tris·HCl, 15 mM NaCl, 60 mM KCl, 0.15 mM spermine and 0.5 mM spermidine] and after centrifugation further resuspended in MNase digestion buffer containing 2 mM CaCl$_2$. From the resuspended solutions, 100 µl aliquots were taken and incubated with or without MNase (100 units) (New England Biolabs) for 5 min. The reaction was stopped by adding 80 µl of MNase digestion buffer, 20 µl of MNase stop buffer (100 mM EDTA, 10 mM EGTA), 3 µl of proteinase K (25 mg/ml) (Roche), and 10 µl of 20% SDS, and then incubated overnight at 37 °C. This was followed by phenol–chloroform extraction and ethanol precipitation. The purified DNA was subjected to qPCR analysis using the primer sets described for ChIP-qPCR analysis.

**Cell lines and culture conditions.** All cell lines were obtained from ATCC. PC3 and DU145 cells were cultured in F-12K medium and Eagle's minimum essential medium supplemented with fetal bovine serum (FBS) to a final concentration of 10%, respectively. C4-2B cells were cultured in Dulbecco's modified Eagle's medium (DMEM)/F12 (4:1) supplemented with heat-inactivated FBS (to 10%) and T-medium supplement (to 1×). To prepare 10×T-medium supplement in 100 ml of 0.1% BSA (Sigma cat# A8022) in PBS, the following components were added: 50 mg insulin (Gibco cat# 12585-014), 136 ng triiodo-L-thyronine (Sigma cat# T2877), 50 mg transferrin (Sigma cat# T4382), 2.5 mg D-Biotin (Sigma cat# 47868), and 250 mg adenine (Sigma cat# A3159). RWPE-1 and PZ-HPV-7 cells were maintained in keratinocyte serum free medium (K-SFM, Gibco 17005-042), supplemented with 0.05 mg/ml of bovine pituitary extract and 5 ng/ml of human recombinant epidermal growth factor. HPrEC cells were maintained in prostate epithelial cell basal medium (ATCC, PCS-440-030) supplemented with prostate epithelial cell growth kit (ATCC, PCS-440-040). HeLa cells were cultured in DMEM supplemented with 5% FBS.

**Plasmids.** The mammalian expression plasmid for ARID4B was described previously[42]. The mammalian expression plasmids for HA-PTEN and HA-PTEN G129R were purchased from Addgene. For luciferase reporter gene assays, the promoters of human *ARID4B* (−2324 to +125 bp), *PIK3CA* (−1448 to +1099 bp), and *PIK3R2* (−1303 to +1324 bp) were amplified by PCR using genomic DNA prepared from PC3 cells and cloned into the pGL3-basic vector (Promega). For PCR amplification of the promoters, the following primers were used:
*ARID4B*:
*ARID4B-Xho*I-PF2: 5′-CCGCTCGAGCGTATACAGTGCTAGCACAGAGTC-3′
*ARID4B-Hind*III-PR2: 5′-CCCAAGCTTCTAGGGTCAGATTACCTGACTTGC-3′
*PIK3CA*:
*PIK3CA-Sac*I-PF4: 5′-CGAGCTCCGACCACAAAGAGGAACAGATCC-3′
*PIK3CA-Bgl*II-PR4: 5′-GGAAGATCTCTGGAATGAGGAAGCTTCACCAAG-3′
*PIK3R2*:
*PIK3R2-Kpn*I-PF12: 5′-CGGGGTACCCTGCGAGTACTCCATCATAGCC-3′
*PIK3R2-Nhe*I-PR13: 5′-CTAGCTAGCCCTCTATCTCCAGCAGTCACAG-3′

**Transfection and luciferase reporter gene assay.** Plasmid transfection into DU145 cells was carried out by FuGene HD (Promega) according to the

manufacturer's instructions. Plasmid transfection into HeLa cells was accomplished using Lipofectamine 2000™ Transfection Reagent (Invitrogen) according to the manufacturers' instructions. Forty-eight hours after transfection, whole cell lysates were prepared and luciferase activity was determined by the luciferase assay system following the manufacturer's instructions (Promega).

Transfection of SiRNA was carried out using Lipofectamine™ RNAiMAX Transfection Reagent (Invitrogen) according to the manufacturers' instructions. Seventy-two hours after transfection, total RNA was purified for RNA-Seq and qRT-PCR analysis, or whole cell lysates were prepared for luciferase reporter gene assays. MISSION® Predesigned siRNA targeting human *ARID4B* (SASI_Hs01_00134230 and SASI_Hs02_00349366) and human *PTEN* (SASI_Hs01_00196478 and SASI_Hs01_00196479) were purchased from Sigma-Aldrich.

To re-express PTEN in PC3 cells, we generated stable PC3 clones with doxycycline-inducible expression of PTEN using the T-REx system according to the manufacturer's instructions (Invitrogen). First, PC3 cells were transfected with pLenti6/TR (Invitrogen) using Lipofectamine 2000 (Invitrogen), and selected by blasticidin. The PC3-TR clones stably expressing the Tet repressor was confirmed by western blot analysis. The human *PTEN* cDNA was ligated into the pcDNA4/TO/Myc-His A vector (Invitrogen) to generate pCDNA4/TO-PTEN. PC3-TR cells were then transfected with the pcDNA4-TO-PTEN vector and were selected by zeocin. Upon addition of doxycycline, stable clones with inducible expression of PTEN were confirmed by western blot analysis.

**Knockout using CRISPR.** To generate the knockout of *ARID4B* in PC3 and DU145 cell lines (*ARID4B*KO), the plasmids with *ARID4B* sgRNA were transfected into cells using Lipofectamine 2000. The ARID4B CRISPR guide RNA plasmid targeting human *ARID4B* was purchased from GenScript. These two *ARID4B* sgRNA sequences are ATCAGTGCCCACTGTCAAAT and AGTTCAGGATGACCACA-TAA. The level of ARID4B protein in each *ARID4B*KO clone was determined by western blot analysis. Cell proliferation assays were performed using Cell Proliferation Kit I (MTT) (Roche) according to the manufacturer's instructions.

**Xenograft prostate cancer model.** The in vivo tumor growth of *ARID4B*KO PC3, *ARID4B*KO DU145, and their control cells was determined using subcutaneous transplant xenograft models. Cancer cells ($2 \times 10^6$) in PBS/Matrigel mixture (Geltrex™ LDEV-free reduced growth factor basement membrane matrix, Gibco) were injected subcutaneously into 6-week-old male NOD/SCID mice (The Jackson Laboratory) under deep anesthesia. The sizes of tumors were measured twice a week. Once the largest tumor diameter reached 1.5 cm, all mice were sacrificed and tumors were harvested.

**Migration, cell invasion, and tumorsphere formation assays.** Cell migration and cell invasion of the control and *ARID4B*KO PC3 cells were examined using CytoSelect™ Cell Migration (CBA-100) and CytoSelect™ Cell Invasion Assay (CBA-110), respectively, following the manufacturer's instructions. Briefly, cells ($5 \times 10^4$) in serum-free culture medium were seeded into 24-well Transwell® Inserts (8.0 μm) of the kits. Medium containing 10% FBS was added to the remaining receiver wells. After 24 h, the inside of each insert was gently swabbed, and stained with crystal violet solution for 10 min. The migratory and invasive cells were imaged with a light microscope under high magnification objective, and counted. At least three individual fields per insert were counted, and the results were from three independent experiments.

The tumorsphere formation assay was modified from a previously published protocol[43]. Briefly, the control and *ARID4B*KO PC3 cells were washed twice in 1× PBS, trypsinized, centrifuged and resuspended in serum-free DMEM-F12 to obtain a single cell suspension. Then, cells ($1 \times 10^4$) were seeded in six-well ultra-low attachment culture plates (Corning) with DMEM-F12 supplemented with 20 ng/μl hEGF (Gibco), 20 ng/μl bFGF (Gibco), 1× B27 without vitamin A (Invitrogen), 1× insulin-transferrin-selenium A (Invitrogen), and 100 U/100 μg/ml penicillin-streptomycin. The plates were incubated in a 5% $CO_2$, humidified incubator at 37 °C for 2 weeks with replenishment of medium every 3 days. Tumorspheres that were 60 μm or larger in size were counted. The results were from three independent experiments performed in triplicate.

**Bioinformatic analyses from human prostate cancer datasets.** Data on genomic alterations of *ARID4B* and *PTEN* from human prostate cancer datasets[2,20–25,34] were acquired from The cBioPortal for Cancer Genomics (http://www.cbioportal.org)[44,45]. Z-scores for mRNA expression of *ARID4B* and *PIK3CA* with the corresponding Pearson correlation coefficient were obtained for prostate cancer samples in the TCGA database. Heat maps with a three-color scale set at −3, 0, and 3 were obtained from the cBioPortal for cancer genomics (http://www.cbioportal.org)[44,45].

As shown in Supplementary Fig. 1d, Kaplan–Meier survival analysis was conducted using the PROGgeneV2 prognostics database[46] to seek correlations between expression of *ARID4B* in prostate tumors and recurrence-free survival in patients from the GSE40272[28] and the ref.[27] datasets. Levels of *ARID4B* expression were used to define high- and low-risk scores. The two groups (high and low expression of *ARID4B*) were generated by splitting patients' data based on the median value of *ARID4B* expression.

**Prognostic index calculation of *PTEN/ARID4B/PIK3CA* signature.** The predictive power of the *PTEN/ARID4B/PIK3CA* three-gene signature for tumor recurrence was calculated from three datasets: the TCGA provisional, the GSE21032[2], and the GSE40272[28] datasets. The prognostic index (PI) was used to define high- and low-risk scores, and the two risk groups (high and low) were generated by splitting patients' data at the median PI. The PI of multiple genes was calculated as $PI = x1\beta1 + x2\beta2 + \ldots + xp\beta p$, where $x$ is the gene expression value and $\beta$ is a risk coefficient obtained from the Cox fitting. The fitting was performed in R (http://cran.r-project.org) using the survival package. The $\beta$ represents the correlation between gene expression levels and the number of events (in our case, death from cancer recurrence). A positive $\beta$ means a higher gene expression level may contribute to a higher death rate, and a negative $\beta$ means a higher gene expression value may contribute to a lower death rate. In our case, *PTEN* has a negative $\beta$ coefficient, while *ARID4B* and *PIK3CA* both have positive $\beta$ coefficients. Patients with lower *PTEN* expression levels or with higher *ARID4B* and/or *PIK3CA* levels would have higher calculated PIs, and therefore would be classified into the high-risk group.

The multivariate Cox proportional hazards model of the *ARID4B/PTEN/PIK3CA* signature combined expression levels of these three genes with time to recurrence in prostate cancer patients from the TCGA provisional dataset using statistical software SPSS19.0, and from the GSE21032[2] and GSE40272[28] datasets using the SurvExpress web tool[47] (http://bioinformatica.mty.itesm.mx:8080/Biomatec/SurvivaX.jsp). In addition, Kaplan−Meier survival curves were used to test whether the *ARID4B/PIK3CA/PTEN* signature distinguishes between high- and low-risk groups for the prostate cancer recurrence in the TCGA provisional ($P = 0.025$), GSE21032[2] ($P = 0.002$), and GSE40272[28] ($P = 0.006$) datasets. Statistical analysis: log-rank test.

**Statistical analysis.** All mouse experiments were performed using 3–16 mice. All experiments from cell lines were performed using at least three independent experiments in triplicate. Means were calculated from at least three independent experiments. All results were shown as the means ± standard errors of the means (SEM). Two-tailed unpaired Student's $t$ test, log-rank test, or Wald chi-squared test were used for statistical analyses. $P$ values less than 0.05 were considered to be statistically significant.

**Reporting summary.** Further information on research design is available in the Nature Research Reporting Summary linked to this article.

## Data availability

RNA-Seq and ChIP-Seq raw data that support our findings of this study were deposited in the Gene Expression Omnibus (https://www.ncbi.nlm.nih.gov/geo/) with the accession code GSE116670. All other data in this study are available from the corresponding author upon reasonable request. A reporting summary for this article is available as a Supplementary Information file. The source data underlying Figs. 1a, b, d, 2c–e, 3b–e, 4d, 5a, b, d–j, 6a–c and Supplementary Figs. 1c, 2b, 3a, 3b, 4c, 4d, 5a–c, 6a–d, 10a–c, 12a–c, 13a–c, 14a, b, 15a, b, 16b, c are provided as a Source Data file.

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

## Acknowledgements

We thank Drs. Sophia Y. Tsai (Baylor College of Medicine) and Lopa Mishra (George Washington University) for comments and critical reading of the manuscript. This work was supported by grants R01CA188471 and R21CA187857 to R.-C.W. from the National Institutes of Health, and Katzen Cancer Research Center to R.-C.W., McCormick Genomic and Proteomic Center to R.-C.W. and M.-Y.W., and a startup fund to M.-Y.W. from George Washington University.

## Author contributions

M.-Y.W., R.-C.W., and I.-C.Y. performed most of the experiments. I.-C.Y. and M.-Y.W. performed bioinformatics analyses from human prostate cancer datasets. Y.-F.C. performed experiments on mouse samples. S.-T.C. performed a plasmid cloning experiment. A.T. analyzed histological data. M.-Y.W and R.-C.W. provided guidance with experimental design and wrote the manuscript.

## Additional information

**Competing interests:** The authors declare no competing interests.

