## [Peer Review File · Nature Communications]

Reviewers' comments:

Reviewer #1, Expertise: prostate cancer, PI3K signalling (Remarks to the Author):

The current manuscript evaluates the relationship between the tumor suppressor PTEN and ARID4B, a chromatin remodeling factor during prostate tumorigenesis.

The authors have studied the role of ARID4B both in vitro and in vivo using immortalized human prostate cancer cell lines (namely PC3 and DU145) and a transgenic mouse model of prostate cancer driven by PTEN deletion respectively, and report the frequency of ARID4B genetic alterations in prostate cancer patient genomic datasets.

The major observation for this study is that ARID4B is a transcriptional activator of PIK3CA and PIK3R2, and that ARID4B deletion partially rescues the Pten-deleted PCa prostate cancer phenotype in mice. These observations are interesting for the field and provide further insight into the tumor suppressive function of PTEN.

The authors also report that a PTEN/PIK3CA/ARID4B gene signature is predictive of prostate cancer patient outcome, although to this reviewer the data do not merit this claim.

Specific points:

1. The authors should examine components of the PI3K pathway by WB as in Fig 1e for PC3 and DU145 cells with ARID4B KD relative to control.
2. It is not clear whether ARID4B plays a role during normal prostate homeostasis. The authors should report whether ARID4B deletion in mouse prostate causes a prostate phenotype, providing further insight into how ARID4B loss reduces PTEN-deleted prostate tumorigenesis.
3. The authors report that a PTEN/ARID4B/PIK3CA gene signature is predictive for PSA-recurrence free survival in patients with prostate cancer, however this reviewer does not believe the data currently merits this statement. The methodology of how the gene signature was generated is missing and needs to be included. Were ARID4B gene amplifications/mutations selected for, or were patients with ARID4B homozygous deletion also included?
4. Is the observed mutual exclusivity for ARID4B amplification/mutation and PTEN loss/mutation in the prostate cancer datasets assessed in Fig 3A & S1B statistically significant? Are the ARID4B mutations detected in patients with prostate cancer activating mutations? Simple statistical analysis of observed vs expected coincidence frequencies would help.
5. Given that tumor burden was significantly reduced upon ARID4B KD in DU145 that retain PTEN expression suggests that ARID4B plays an oncogenic role that is not dependent on PTEN as the authors claim. The authors should determine if PTEN status and PI3K signalling activation is altered upon ARID4B loss in DU145 cells with ARID4B KD relative to the control. The authors should discuss why a response was observed in DU145 cells
6. The authors should determine whether PC3 and DU145 cells over-express ARID4B relative to non-malignant prostate cell lines.
7. Fig S1 b/c indicate the predictive value of ARID4B expression in prostate cancer patients, however the sample size for the datasets shown are small. The authors should confirm the smaller datasets reported have sufficient statistical power to detect a true positive
8. Does ARID4B loss delay tumour onset in addition to causing slower progression? Perhaps the authors could consider analyzing an earlier, pre-malignant time-point.
9. H&E images in Fig S5c need to be revised to accurately show the DLP glands in control and

PtenPC^{-/-};Aridb4PC^{-/-} mice.

10. The authors should further explore the mechanism whereby ARID4B loss reduces PtenPC^{-/-} tumor burden by analyzing apoptosis/senescence e.g. WB for cleaved-caspase-3, p53, p21 and/or p27.

Reviewer #2, Expertise: chromatin remodelling (Remarks to the Author):

In this manuscript, the authors present data implicating ARID4B as a necessary intermediate for PTEN regulation of PI3K signaling. The authors use cell culture, genetically-engineered mouse models and bioinformatic studies to support this conclusion. If correct, inhibitors of ARID4B activity could provide an effective therapeutic approach for PTEN-deficient human prostate cancer. While the genetically-engineered mouse studies clearly support the overall hypothesis of the authors, the cell culture studies and, to a limited extent, the bioinformatic mining of primary human prostate cancer sequencing data are not sufficiently rigorous. Without these key data, the studies do not provide any mechanistic insights into how ARID4B loss leads to the tumor inhibition observed in the mouse studies. Therefore, without high quality data from the cell culture studies, a manuscript focused upon the genetically-engineered mouse studies appear well-suited for a specialty journal.

Major Points:

- 1) One of the major concerns about this study comes from the authors' model that ARID4B essentially functions in a vacuum. They acknowledge that it is a component of the mSIN3A histone deacetylase complex yet seem surprised that it lacks a methyltransferase or acetyltransferase activity. As a component of the mSIN3A complex, it presumably regulates the activity of the histone deacetylase HDAC1, similar to how other components of chromatin remodeling complexes work. They provide no studies to address the effects of ARID4B on the mSIN3A complex and how that might influence their results.
- 2) The majority of the bioinformatics results are correlative rather than showing evidence of causation.
- 3) The vast majority of the cell culture studies rely on one prostate cancer cell line- PC3. The authors do use one additional prostate cancer cell line as a negative control but additional cell lines are needed.
- 4) The gene expression changes after knockdown of ARID4B in the PC3 cell line appear modest with no supporting Western blot data (Figure 1A-B).
- 5) The authors use transient expression of luciferase reporters to measure the effect of ARID4B on gene transcription. However, the authors propose that ARID4B works as a chromatin remodeler. Therefore, the luciferase experiments do not provide an appropriate model for their studies because plasmid DNA does not undergo chromatinization 48 hours after transfection.
- 6) The ChIP-seq data in Figures 2a&b and SuppFigure3A are problematic. The similarity between the signals, especially in the heat map, between the ARID4B and Input DNA suggest that the authors may have sheared the DNA too much. If so, they would tend to find their signals around TSS. It appears that they used Active Motif to carry out the bioinformatics analysis. They should enlist the expertise of an academic bioinformatician to re-analyze these data. The ChIP-pcr data are more convincing of recruitment of ARID4B to the promoters of PIK3CA and PIK3R2. However, showing that ARID4B is absent from the MYOD1 promoter is not sufficient to establish the specificity of ARID4B binding throughout the genome. Perhaps ARID4B is present at all actively

transcribed genes. Did the authors integrate the RNA-seq data to the ChIP-seq data to address this association?

7) The authors observe a negative correlation between the presence of ARID4B and histone H1 at 3 promoters and suggest that this provides the mechanism for how ARID4B activates expression of PIK3CA and PIK3R2. Their limited data provide a tenuous connection. To support this notion, they need to compare H1 ChIP-seq to ARID4B ChIP-seq in the presence or absence of ARID4B expression. It was also not clear why they used H3K4me3 as a control for these experiments. Considering that ARID4B is a component of a HDAC complex, wouldn't it have made more sense to examine H3 and H4 acetylation?

8) The authors do not provide any statistical analyses for the primary tumor data in Figure 3A. Furthermore, the rest of the data in this figure demonstrate that loss of ARID4D almost completely abrogates growth of the PC3 cell line. Therefore, the fact that the cells don't grow in tumorspheres or in mice doesn't offer much insight into its mechanism of action.

9) The authors present no comparison of the 3 gene signature as a predictor for prognosis of prostate cancer recurrence to other available prognostic signatures.

10) The authors do not present key controls such as- does reexpression of ARID4B rescue the growth effects observed in the knockout cell line and does the absence of ARID4B prevent increased PIK3CA and PIK3R2 expression upon PTEN reexpression?

11) The Discussion is far too long, mainly due to the authors' repetition of the results. The authors should focus on interpreting the basic science and translational implications of their results.

Subject: Manuscript NCOMMS-18-28842.

Reviewers' comments:

Reviewer #1, Expertise: prostate cancer, PI3K signalling (Remarks to the Author):

The current manuscript evaluates the relationship between the tumor suppressor PTEN and ARID4B, a chromatin remodeling factor during prostate tumorigenesis.

The authors have studied the role of ARIDB4 both in vitro and in vivo using immortalized human prostate cancer cell lines (namely PC3 and DU145) and a transgenic mouse model of prostate cancer driven by PTEN deletion respectively, and report the frequency of ARIDB4 genetic alterations in prostate cancer patient genomic datasets.

The major observation for this study is that ARIDB4 is a transcriptional activator of PIK3CA and PIK3R2, and that ARIDB4 deletion partially rescues the Pten-deleted PCa prostate cancer phenotype in mice. These observations are interesting for the field and provide further insight into the tumor suppressive function of PTEN.

The authors also report that a PTEN/PIK3CA/ARIDB4 gene signature is predictive of prostate cancer patient outcome, although to this reviewer the data do not merit this claim.

Specific points:

Q1. The authors should examine components of the PI3K pathway by WB as in Fig 1e for PC3 and DU145 cells with ARID4B KD relative to control.

A1: In the revised manuscript, we showed that knockdown of *ARID4B* in human prostate cancer cell lines PC3, DU145, and LNCaP resulted in reduced expression or phosphorylation of the core regulators and downstream effectors of the PI3K-AKT pathway by western blot analyses (Supplementary Fig. 2a-c).

Q2. It is not clear whether ARID4B plays a role during normal prostate homeostasis. The authors should report whether ARID4B deletion in mouse prostate causes a prostate phenotype, providing further insight into how ARIDB4 loss reduces PTEN-deleted prostate tumorigenesis.

A2: In Supplementary Fig. 9a and 9b, we showed that mice carrying prostate-specific *Arid4b* deletion (*Arid4b*^{PC^{-/-}}) exhibited normal prostate size and histology.

Q3-1. The authors report that a PTEN/ARID4B/PIK3CA gene signature is predictive for PSA-recurrence free survival in patients with prostate cancer, however this reviewer does not believe the data currently merits this statement. The methodology of how the gene signature was generated is missing and needs to be included. Were ARIDB4 gene amplifications/mutations selected for, or were patients with ARIDB4 homozygous deletion also included?

A3-1: In the revised manuscript, we included the derivation of the *ARID4B/PIK3CA/PTEN* three-gene signature in the Methods section and Supplementary Fig 12 as follows.

“The multivariate Cox proportional hazards model of the *ARID4B/PTEN/PIK3CA* signature combines these three-gene expression levels with other clinical and pathological variables into an integrated risk score model to determine the correlations of *ARID4B*, *PTEN*, and *PIK3CA* with time to recurrence in prostate cancer patients from the two datasets GSE40272 (ref. 1) and GSE21032 (ref. 2) using the SurvExpress web tool (<http://bioinformatica.mty.itesm.mx:8080/Biomatec/SurvivaX.jsp>). Briefly, the prognostic index (PI) was used to define and generate high and low risk groups with classical multivariate Cox model. The PI was estimated by explanatory variables including selected the gene expression level, patient age, Gleason score, PSA level, prostate cancer stage, and follow-up status. The Cox fitting was performed in R using the *survival* package. Higher PI represents higher risk, and the two risk groups (high and low) were generated by splitting patients with the median PI. Our results revealed an increased predictive power of the *PTEN/ARID4B/PIK3CA* three-gene combinatorial signature for tumor recurrence over that of individual gene alone or any two genes combined (Fig. 5h and Supplementary Fig. 12).”

Single gene validation for recurrence-free survival (RFS)

Gene	Hazard ratio	P value
ARID4B	2.34	0.059
PIK3CA	0.90	0.790
PTEN	1.44	0.379

Two genes-combined validation for recurrence-free survival

Gene 1	Gene 2	Hazard ratio	P value
ARID4B +	PIK3CA	2.41	0.050
ARID4B +	PTEN	4.11	0.010
PIK3CA +	PTEN	1.16	0.720

Three gene-combined validation for recurrence-free survival

Gene 1	Gene 2	Gene 3	Hazard ratio	P value
ARID4B +	PIK3CA +	PTEN	4.58	0.006

Supplementary Fig. 12 Validation of the *ARID4B/PIK3CA/PTEN* three-gene signature for prostate cancer recurrence using the GSE40272 dataset. The prognostic index (PI), also known as the risk score, is used to generate risk groups with classical multivariate Cox model. PI is defined and estimated by explanatory variables including gene expression level, age, Gleason score, PSA level, prostate cancer stage and follow-up status with the SurvExpress web tool. The Cox fitting is performed in R using the *survival* package. Higher PI represents higher risk, and the two risk groups (high and low) were generated by splitting patients with the median PI. When the patient PI was estimated with single gene (*ARID4B*,

PIK3CA or *PTEN*) and corresponding explanatory variables, non-significant correlation with patient prognosis was found (*ARID4B*: HR: 2.34, $P = 0.059$; *PIK3CA*: HR: 0.9, $P = 0.79$; *PTEN*: HR: 1.44, $P = 0.379$). On the other hand, combination of *ARID4B* with *PIK3CA* or *PTEN* (combination of two genes) increases the hazard ratio with significant difference between high- and low-risk groups (*ARID4B* + *PIK3CA*: HR: 2.41, $P = 0.05$; *ARID4B* + *PTEN*: HR: 4.11, $P = 0.01$). Furthermore, the *PTEN/ARID4B/PIK3CA* three-gene combinatorial signature presents the most predictive power for tumor recurrence (HR: 4.58, $P = 0.006$) over that of individual gene alone or any two genes combined.

In addition, the Kaplan-Meier survival curve was used as a platform to show that the *ARID4B/PIK3CA/PTEN* signature could distinguish the high-risk group from the low-risk group for the prostate cancer recurrence. According to this *ARID4B/PTEN/PIK3CA* three-gene signature, patients in high-risk group have a lower survival rate than those in the low-risk group from the two datasets GSE40272 (ref. 1) and GSE21032 (ref. 2) (Fig 5i, hazard ratio 4.58, $P = 0.006$; Supplementary Fig. 13b, hazard ratio 3.23, $P = 0.002$). These results suggest that the *ARID4B/PIK3CA/PTEN* three-gene signature possesses the potential as a prognosis marker for PSA recurrence in patients with prostate cancer.

In the Discussion section of the revised manuscript, we compared the *PTEN/ARID4B/PIK3CA* three gene signature with the published *PTEN/SMAD4/CCND1/SPP1* four gene signature (ref. 3) as follows.

“Similar to the *PTEN/ARID4B/PIK3CA* three gene signature, it has been reported that *PTEN* together with *SMAD*, *CCND1*, and *SPP1* form a four gene signature that is prognostic of recurrence and lethal metastasis in human prostate cancer (ref. 3). *SMAD4* in the $TGF\beta$ pathway is a suppressor of prostate tumor progression, whereas cyclin D1 and *SPP1* act as mediators of the prostate cancer processes. The inactivation of *PTEN* and *SMAD4* as well as activation of cyclin D1 and *SPP1* promote prostate cancer. Yet, combination of *PTEN/SMAD4* do not by themselves predict risk for prostate cancer recurrence. However, when combined with Cyclin D1 and *SPP1*, the prognostic value of the *PTEN/SMAD4/CCND1/SPP1* four gene signature were enhanced,

providing significant prediction for prostate cancer recurrence risk. This four-gene model distinguished the high-risk patient group from the low-risk group for the prostate cancer recurrence by Kaplan–Meier analysis [hazard ratio (HR) = 2.6, $P = 0.012$]. Similarly, we showed the *PTEN/ARID4B/PIK3CA* three-gene combinatorial signature has the most predictive power for tumor recurrence over that of individual gene alone or any two genes combined. This *PTEN/ARID4B/PIK3CA* signature robustly stratified patients into high- and low-risk groups for tumor recurrence (HR = 4.58, $P = 0.006$ in the GSE40272 dataset, and HR = 3.23, $P = 0.002$ in the GSE21032 dataset). In addition, the *PTEN/SMAD4/CCND1/SPP1* four gene signature involves in the PTEN and TGF β pathways that converge to regulate prostate cancer progression (ref. 3). On the other hand, we identified the PTEN-ARID4B-PI3K regulatory axis that illustrates the molecular mechanism underlying the predictive power of the *PTEN/ARID4B/PIK3CA* signature for tumor recurrence in prostate cancer patients.”

References

1. Gulzar ZG, McKenney JK, Brooks JD. Increased expression of NuSAP in recurrent prostate cancer is mediated by E2F1. *Oncogene* **32**, 70-77 (2013).
2. Taylor BS, *et al.* Integrative genomic profiling of human prostate cancer. *Cancer Cell* **18**, 11-22 (2010).
3. Ding Z *et al.* SMAD4-dependent barrier constrains prostate cancer growth and metastatic progression, *Nature* 470, 269–276 (2011)

Q3-2. The authors report that a PTEN/ARID4B/PIK3CA gene signature is predictive for PSA-recurrence free survival in patients with prostate cancer, however this reviewer does not believe the data currently merits this statement. The methodology of how the gene signature was generated is missing and needs to be included. **Were ARID4B gene amplifications/mutations selected for, or were patients with ARID4B homozygous deletion also included?**

A3-2: We analyzed the correlations of the *ARID4B/PIK3CA/PTEN* three-gene signature with time to recurrence in prostate cancer patients from the two datasets GSE40272 (ref. 1) (Fig. 5h, i) and GSE21032 (ref. 2) (Supplementary Fig. 13a, b) using the SurvExpress web tool (<http://bioinformatica.mty.itesm.mx:8080/Biomatec/SurvivaX.jsp>).

The SurvExpress web tool includes every samples in the dataset for analysis. In Fig. 5h and 5i, the GSE40272 (ref. 1) dataset (n = 103) contained 1% of patient samples with *ARID4B* amplification that were included in our analysis. In Supplementary Fig. 13a and 13b, the GSE21032 (ref. 2) dataset does not contain the information pertaining to the genome status of samples.

References

1. Gulzar ZG, McKenney JK, Brooks JD. Increased expression of NuSAP in recurrent prostate cancer is mediated by E2F1. *Oncogene* **32**, 70-77 (2013).
2. Taylor BS, *et al.* Integrative genomic profiling of human prostate cancer. *Cancer Cell* **18**, 11-22 (2010).

Q4-1. Is the observed mutual exclusivity for ARID4B amplification/mutation and PTEN loss/mutation in the prostate cancer datasets assessed in Fig 3A & S1B statistically significant? Are the ARID4B mutations detected in patients with prostate cancer activating mutations? Simple statistical analysis of observed vs expected coincidence frequencies would help.

A4-1: Although statistical analysis of the *ARID4B-PTEN* mutual exclusivity in prostate cancer datasets didn't show the significance ($P > 0.05$) (Supplementary Fig. 6), a tendency of mutual exclusivity between *ARID4B* and *PTEN* deletions in prostate cancer genome was observed in all five datasets.

In most cohorts, *ARID4B* gene alteration rate is less than 5%. It is possible that the extremely few cases of *ARID4B* gene alterations render it mathematically difficult to obtain a significant P value. Similar to our findings, a previous report by others also suggests a tendency of mutual exclusivity between *CHDI* and *PTEN* deletions in prostate and breast cancer without reaching statistical significance (ref. 4). It also has been reported that even though the mutual exclusivity between *ARID1B* and *KRAS* is not significant in any individual cancer type, the pair show statistical significance for mutual exclusivity when multiple cancer types were combined to increase the sample number and to reach the statistical power (ref. 5).

Importantly, the tendency of *ARID4B-PTEN* mutual exclusivity led us to further evaluate the intricate relations between these two genes. The extensive *in vitro* and *in vivo* data, including those from depletion of *ARID4B* in the *PTEN*-deficient prostate cancer cell line PC3 and the prostate-specific *ARID4B* and *PTEN* double knockout mouse model, demonstrated the dependency on *ARID4B* by *PTEN*-deficient prostate cancer cells.

We added Supplementary Fig. 6 as follows.

Cohort	Sample size	ARID4B alteration rate (n)	PTEN alteration rate (n)	Tendency	P value
Armenia et al. Nat Genetics 2018	1,013	3% (30)	16 % (162)	Mutual exclusivity	0.500
TCGA, Cell 2015	333	4 % (13)	17 % (56)	Mutual exclusivity	0.599
Robinson et al. Cell 2015	150	2.7 % (4)	40 % (60)	Mutual exclusivity	0.649
Kumar et al. Nat Med 2016	54	9 % (4)	44 % (23)	Mutual exclusivity	0.110
Taylor et al. Cancer Cell 2010	103	1 % (1)	14 % (14)	Mutual exclusivity	0.854

Supplementary Fig. 6 The tendency of mutual exclusivity between *ARID4B* and *PTEN* deletions. Analysis of five prostate cancer genome datasets using the cBioportal data source (<http://www.cbioportal.org/>) suggests a tendency of mutual exclusivity between *ARID4B* and *PTEN* deletions in prostate cancer genome.

In the Discussion section of this revised manuscript, we added the following paragraph.

“Analysis of five prostate cancer genome datasets using the cBioportal data source (<http://www.cbioportal.org/>) suggests a tendency of mutual exclusivity between *ARID4B* and *PTEN* deletions in prostate cancer genome, although it does not research the statistical significance ($P > 0.05$) (Supplementary Fig. 6). In most cohorts, *ARID4B* gene alteration rate is less than 5%. It is possible that the extremely few cases of *ARID4B* gene alterations render it mathematically difficult to obtain a significant P value. Similar to our findings, a previous report by others suggests a tendency of mutual exclusivity between *CHD1* and *PTEN* deletions in prostate and breast cancer without reaching statistical significance (ref. 4). It also has been reported that even though the mutual exclusivity between *ARID1B* and *KRAS* is not significant in any individual cancer type, the pair show statistical significance for mutual exclusivity when multiple cancer types were combined to increase the sample number and to reach the statistical power (ref. 5). Importantly, the tendency of ARID4B-PTEN mutual exclusivity led us to further evaluate the intricate relations between these two genes. The extensive *in vitro* and *in vivo* data, including those from depletion of ARID4B in the PTEN-deficient prostate cancer cell line PC3 and the prostate-specific *ARID4B* and *PTEN* double knockout mouse model, demonstrated the dependency on ARID4B by PTEN-deficient prostate cancer cells.”

References

4. Zhao D *et al.* Synthetic essentiality of chromatin remodeling factor CHD1 in PTEN deficient cancer, *Nature* 542, 484–488 (2017)
5. Kim Y.A. *et al.* MEMCover: integrated analysis of mutual exclusivity and functional network reveals dysregulated pathways across multiple cancer types. *Bioinformatics* 31, i284–i292 (2015)

Q4-2. Is the observed mutual exclusivity for ARID4B amplification/mutation and PTEN loss/mutation in the prostate cancer datasets assessed in Fig 3A & **S1B statistically significant?** Are the ARID4B mutations detected in patients with prostate cancer activating mutations? Simple statistical analysis of observed vs expected coincidence frequencies would help.

A4-2: We have checked the statistical power of the plots with two-tailed two-sample T-Test. With the stated sample number in each plot and the significance level $\alpha = 0.05$, the calculated statistical powers are listed as follows:

Fig. S1b, left: power = 0.841, P value = 0.001.

Fig. S1b, right: power = 0.996, P value < 0.001.

As the statistical powers of all analyses are above desired power level (> 0.800), the sample sizes are sufficient to exclude a false positive interpretation. In addition, the P value of each analysis is less than 0.05, suggesting statistical significance.

Q4-3. Is the observed mutual exclusivity for ARID4B amplification/mutation and PTEN loss/mutation in the prostate cancer datasets assessed in Fig 3A & S1B statistically significant?

Are the ARID4B mutations detected in patients with prostate cancer activating mutations?

Simple statistical analysis of observed vs expected coincidence frequencies would help.

A4-3: Although some *ARID4B* mutations were detected in the prostate cancer genome datasets (Fig. 3a) (refs. 6-10), there is no information about whether these mutations activate or inhibit the *ARID4B* activity. There is also no data suggesting whether these *ARID4B* mutations promote or suppress prostate cancer. In Fig. 3a, all of the *ARID4B* mutations (truncating mutations and missense mutations) were labeled as “unknown significance”.

References

6. Armenia J, *et al.* The long tail of oncogenic drivers in prostate cancer. *Nat Genet* **50**, 645-651 (2018).
7. Cancer Genome Atlas Research N. The Molecular Taxonomy of Primary Prostate Cancer. *Cell* **163**, 1011-1025 (2015).
8. Robinson D, *et al.* Integrative clinical genomics of advanced prostate cancer. *Cell* **161**, 1215-1228 (2015).
9. Kumar A, *et al.* Substantial interindividual and limited intraindividual genomic diversity among tumors from men with metastatic prostate cancer. *Nat Med* **22**, 369-378 (2016).
10. Taylor BS, *et al.* Integrative genomic profiling of human prostate cancer. *Cancer Cell* **18**, 11-22 (2010).

Q5. Given that tumor burden was significantly reduced upon *ARID4B* KD in DU145 that retain *PTEN* expression suggests that *ARID4B* plays an oncogenic role that is not dependent on *PTEN* as the authors claim. The authors should determine if *PTEN* status and *PI3K* signalling activation is altered upon *ARID4B* loss in DU145 cells with *ARID4B* KD relative to the control. The authors should discuss why a response was observed in DU145 cells

A5: In the Results section of this revised manuscript, we included data showing that knockdown of *ARID4B* in the *PTEN*-intact prostate cancer cell line DU145 reduced expression or phosphorylation of the core regulators and downstream effectors of the *PI3K*-*AKT* pathway, whereas expression of *PTEN* remained the same (Supplementary Fig. 2b).

In the Discussion section of the revised manuscript, we added the following paragraph to discuss the possibility why knockout of *ARID4B* in the *PTEN*-intact prostate cancer cell line DU145 also suppressed tumor growth and cell proliferation.

“In both PC3 and DU145 cells, depletion of *ARID4B* resulted in reduced expression or phosphorylation of the core regulators and downstream effectors of the *PI3K*-*AKT* pathway, while expression of *PTEN* in DU145 cells was retained the same (Supplementary Fig. 2a, b). In the *PTEN*-deficient prostate cancer cell line PC3, knockout of *ARID4B* efficiently inhibits cancer cell growth and tumor formation in xenograft, suggesting the dependency on *ARID4B* by *PTEN*-deficient prostate cancer cells. In the *PTEN*-intact prostate cancer cell line DU145, knockout of *ARID4B* also suppressed cell growth and tumor formation in xenograft, but not to the extent as in PC3 cells. It is possible that *PTEN*-deficient PC3 cells is more dependent on activation of the *PI3K*-*AKT* pathway for tumor growth than the *PTEN*-intact DU145 cells. It also cannot be ruled

out that ARID4B plays an oncogenic role to promote cancer cell activity through additional pathways that are not related to PTEN tumor suppressor function. Immunohistochemistry detected an increase of cleaved caspase 3 (an apoptosis marker) positive cells in prostate of the *Pten*^{PC-/-}*Arid4b*^{PC-/-} mice, suggesting apoptosis is also involved (Supplementary Fig. 17a, b). Consistent with this result, the expression levels of apoptosis regulators *p53* and *Fas* were elevated in the *Pten*^{PC-/-}*Arid4b*^{PC-/-} prostate compared to the *Pten*^{PC-/-} prostate (Supplementary Fig. 17c). For its oncogenic function, ARID4B might directly or indirectly inhibit the apoptosis pathway that is mediated by the P53 tumor suppressor.”

Q6. The authors should determine whether PC3 and DU145 cells over-express ARID4B relative to non-malignant prostate cell lines.

A6: In the revised manuscript, we showed that human prostate cancer cell lines PC3, DU145, LNCaP, and C4-2B express higher levels of ARID4B protein than normal (non-malignant) prostate epithelial cells RWPE-1, PZ-HP-7, and HPrEC by western bolt analysis (Supplementary Fig. 1c).

Q7. Fig S1 b/c indicate the predictive value of ARID4B expression in prostate cancer patients, however the sample size for the datasets shown are small. The authors should confirm the smaller datasets reported have sufficient statistical power to detect a true positive.

A7: We have checked the statistical power of the plots with two-tailed two-sample T-Test. With the stated sample number in each plot and the significance level $\alpha = 0.05$, the calculated statistical powers are listed as follows:

For Fig. S1b, left: power = 0.841, *P* value = 0.001.

Fig. S1b, right: power = 0.996, *P* value < 0.001.

Fig. S1d, left (Fig, S1c in previous version): power = 1.000, *P* value = 0.0006.

Fig. S1d, right (Fig, S1c in previous version): power = 1.000, *P* value = 0.031.

As the statistical powers of all analyses are above desired power level (> 0.800), the sample sizes are sufficient to exclude a false positive interpretation. In addition, the *P* value of each analysis is less than 0.05, suggesting statistical significance.

Q8. Does ARID4B loss delay tumour onset in addition to causing slower progression? Perhaps the authors could consider analyzing an earlier, pre-malignant time-point.

A8: We showed that most of the *Pten*^{PC-/-} mice developed high-grade prostatic intraepithelial neoplasia (PIN) or adenocarcinoma at 5 months of age (Fig. 4b, c and Supplementary Fig. 8c). When combined with loss of ARID4B, tumor development in most of the *Pten*^{PC-/-}*Arid4b*^{PC-/-} prostates was arrested at the normal/hyperplastic (40%) or PIN (60%) stage at 5 months of age (Fig. 4b, c and Supplementary Fig. 8c), suggesting that ablation of ARID4B attenuated prostate

cancer progression. At earlier time points (7 weeks and 3 months of age), the *Pten*^{PC-/-} *Arid4b*^{PC-/-} mice did not develop prostate cancer (data not shown).

To analyze whether ablation of ARID4B delays the onset of prostate cancer caused by PTEN deficiency, we examined the *Pten*^{PC-/-} *Arid4b*^{PC-/-} mice at later time point (9 months of age). While the *Pten*^{PC-/-} mice exhibited enlarged prostates, the *Pten*^{PC-/-} *Arid4b*^{PC-/-} mice showed comparable prostate size with the control mice (Supplementary Fig. 10a). Similar to the results from 5 months-old mice, only normal/hyperplasia (40%) and PIN (60%) were found in the *Pten*^{PC-/-} *Arid4b*^{PC-/-} mice (Supplementary Fig. 10b, c). These results suggest that the PIN in the *Pten*^{PC-/-} *Arid4b*^{PC-/-} mice did not progress to adenocarcinoma after the extended period. These new data were included in the Results section.

Due to the tumor burden, most of the *Pten*^{PC-/-} mice were very sick at 9 months of age, we cannot keep these mice beyond 9 months.

Q9. H&E images in Fig S5c need to be revised to accurately show the DLP glands in control and *Pten*^{PC-/-}; *Arid4b*^{PC-/-} mice.

A9: In Supplementary Fig. 8c of the revised manuscript (Fig. S5c in previous version), new H&E images of the DP glands from the control and *Pten*^{PC-/-} *Arid4b*^{PC-/-} mice were used.

Q10. The authors should further explore the mechanism whereby ARID4B loss reduces *Pten*^{PC-/-} tumor burden by analyzing apoptosis/senescence e.g. WB for cleaved-caspase-3, p53, p21 and/or p27.

A10: In this revised manuscript, we analyzed apoptosis in prostates of the *Pten*^{PC-/-} and *Pten*^{PC-/-} *Arid4b*^{PC-/-} mice by immunohistochemistry staining of cleaved caspase 3 (an apoptosis marker). In addition, we compared the levels of *p53* and *Fas* expression between the *Pten*^{PC-/-}, and *Pten*^{PC-/-} *Arid4b*^{PC-/-} prostates by qRT-PCR. We added the following to the Discussion section.

“Immunohistochemistry detected an increase of cleaved caspase 3 positive cells in the *Pten*^{PC-/-} *Arid4b*^{PC-/-} prostate compared to the *Pten*^{PC-/-} prostate, suggesting that ablation of ARID4B promoted apoptosis (Supplementary Fig. 17a, b). Consistent with this result, the expression levels of apoptosis regulators *p53* and *Fas* were elevated in the *Pten*^{PC-/-} *Arid4b*^{PC-/-} prostate (Supplementary Fig. 17c). For its oncogenic function, ARID4B might directly or indirectly inhibit the apoptosis pathway that is mediated by the P53 tumor suppressor.”

Reviewer #2, Expertise: chromatin remodelling (Remarks to the Author):

In this manuscript, the authors present data implicating ARID4B as a necessary intermediate for PTEN regulation of PI3K signaling. The authors use cell culture, genetically-engineered mouse models and bioinformatic studies to support this conclusion. If correct, inhibitors of ARID4B

activity could provide an effective therapeutic approach for PTEN-deficient human prostate cancer. While the genetically-engineered mouse studies clearly support the overall hypothesis of the authors, the cell culture studies and, to a limited extent, the bioinformatic mining of primary human prostate cancer sequencing data are not sufficiently rigorous. Without these key data, the studies do not provide any mechanistic insights into how ARID4B loss leads to the tumor inhibition observed the mouse studies. Therefore, without high quality data from the cell culture studies, a manuscript focused upon the genetically-engineered mouse studies appear well-suited for a specialty journal.

Major Points:

Q1) One of the major concerns about this study comes from the authors' model that ARID4B essentially functions in a vacuum. They acknowledge that it is a component of the mSIN3A histone deacetylase complex yet seem surprised that it lacks a methyltransferase or acetyltransferase activity. As a component of the mSIN3A complex, it presumably regulates the activity of the histone deacetylase HDAC1, similar to how other components of chromatin remodeling complexes work. They provide no studies to address the effects of ARID4B on the mSIN3A complex and how that might influence their results.

A1: In the revised manuscript, we showed that knockdown of HDAC1 by two different short interfering RNAs *SiHDAC1-1* and *SiHDAC1-2* resulted in increased expression of *PIK3CA* and *PIK3R2* in prostate cancer cell lines PC3, DU145, and LNCaP (Supplementary Fig. 14a-c). These results suggest that HDAC1 suppressed expression of *PIK3CA* and *PIK3R2*. On the other hand, we showed that knockdown of ARID4B decreased expression of *PIK3CA* and *PIK3R2* in prostate cancer cell lines PC3, DU145, and LNCaP (Supplementary Fig. 2a-c). Our results suggest that ARID4B activated expression of *PIK3CA* and *PIK3R2* in prostate cancer cell lines PC3, DU145, and LNCaP (Fig 1a, b, and Supplementary Fig. 2a-c). Our results indicate the opposing effects of ARID4B and HDAC1 on regulation of *PIK3CA* and *PIK3R2* expression.

As a repressor, HDAC1 deacetylates histones H4 and H3 to inhibit the transcriptionally active states of genes. In this regard, we found that ablation of ARID4B in PC3 cells did not affect the levels of H4 acetylation on the promoters of *PIK3CA* and *PIK3R2* (Supplementary Fig. 5a), suggesting that ARID4B does not mediate the HDAC1 activity on these two promoters.

Together, our results suggest that ARID4B and HDAC1 are likely to function independently of each other to regulate *PIK3CA* and *PIK3R2* in prostate cancer cells. Therefore, the SIN3A/HDAC1 repressor complex does not contribute to activation of *PIK3CA* and *PIK3R2* by ARID4B.

In the revised manuscript, we included the following paragraph in the Discussion section.

“Although ARID4B does not have intrinsic methyltransferase nor acetyltransferase activities, ARID4B contains a Tudor domain and a chromo domain (ref. 11). These two domains may promote assembly of chromatin remodeling complexes (refs. 12, 13). It has been reported that ARID4B is a component of the SIN3A/HDAC1 repressor complex that inhibits expression of target genes (ref. 14). In this report, we showed that ARID4B was recruited to the promoters of *PIK3CA* and *PIK3R2* and activated their expression, suggesting that ARID4B functions as an

activator to up-regulate *PIK3CA* and *PIK3R2*. These results are consistent with our previously findings that ARID4B functions as co-activator (not a co-repressor) for androgen receptor in the testis and is essential for testis development (ref. 15). While knockdown of *ARID4B* decreased expression of *PIK3CA* and *PIK3R2* in prostate cancer cell lines PC3, DU145, and LNCaP (Supplementary Fig. 2), we showed that knockdown of *HDAC1* resulted in increased expression of *PIK3CA* and *PIK3R2* (Supplementary Fig. 14). These results indicate the opposing effects between ARID4B and HDAC1 on regulation of *PIK3CA* and *PIK3R2* expression. As a repressor, HDAC1 deacetylates histones H4 and H3 to inhibit the transcriptionally active states of genes. While HDAC1 suppresses expression of *PIK3CA* and *PIK3R2*, we found that ablation of ARID4B in PC3 cells did not affect the levels of H4 acetylation on the promoters of *PIK3CA* and *PIK3R2* (Supplementary Fig. 5a), suggesting that ARID4B does not mediate the HDAC1 activity on these two promoters. Together, our results suggest that ARID4B and HDAC1 are likely to function independently of each other to regulate *PIK3CA* and *PIK3R2* in prostate cancer cells. Therefore, the SIN3A/HDAC1 repressor complex does not contribute to activation of *PIK3CA* and *PIK3R2* by ARID4B.”

References

11. Lin C, *et al.* Recent advances in the ARID family: focusing on roles in human cancer. *Oncotargets Ther* **7**, 315-324 (2014).
12. Lu R, Wang GG. Tudor: a versatile family of histone methylation 'readers'. *Trends Biochem Sci* **38**, 546-555 (2013).
13. Nishibuchi G, Nakayama J. Biochemical and structural properties of heterochromatin protein 1: understanding its role in chromatin assembly. *J Biochem* **156**, 11-20 (2014).
14. Fleischer TC, Yun UJ, Ayer DE. Identification and characterization of three new components of the mSin3A corepressor complex. *Mol Cell Biol* **23**, 3456-3467 (2003).
15. Wu RC, Zeng Y, Pan IW, Wu MY. Androgen Receptor Coactivator ARID4B Is Required for the Function of Sertoli Cells in Spermatogenesis. *Mol Endocrinol* **29**, 1334-1346 (2015).

Q2) The majority of the bioinformatics results are correlative rather than showing evidence of causation.

A2: Although the bioinformatics results are correlative rather than showing evidence of causation, it aids in identifying *ARID4B* as a cancer-related gene and predicting the role of ARID4B in prostate cancer development. With the information from bioinformatics analyses, we provide a large body of evidence (summarized below) to show causation.

First, in support of bioinformatics analyses which showed that expression of *ARID4B* positively correlates with prostate cancer development (Supplementary Fig. 1), we found that expression of ARID4B in prostate cancer cells is more abundant compared to non-malignant prostate epithelial cells (Supplementary Fig. 1c). In addition, we demonstrated that ARID4B positively regulates the expression of *PIK3CA* and *PIK3R2*, and identified activation of the PI3K/AKT signaling pathway

as the molecular mechanism by which ARID4B is involved in prostate cancer progression (Fig. 1).

Second, in support of the bioinformatics analyses which suggest a tendency of mutual exclusivity between *ARID4B* and *PTEN* deletions in prostate cancer genome (Fig. 3 and Supplementary Fig. 6), we showed the dependency on ARID4B by PTEN-deficient prostate cancer using prostate cancer cell lines and mouse models (Figs. 3, 4).

Third, upon identifying the novel PTEN-ARID4B-PI3K regulatory axis, bioinformatics analyses suggest that the *PTEN/ARID4B/PIK3CA* three-gene signature improves the predictive power for prostate cancer recurrence in patients (Fig. 5h, i and Supplementary Fig. 13a, b).

Q3) The vast majority of the cell culture studies rely on one prostate cancer cell line- PC3. The authors do use one additional prostate cancer cell line as a negative control but additional cell lines are needed.

A3: In the revised manuscript, we used additional cell lines and included more cell culture data to support our findings.

1. We showed that expression of ARID4B in different prostate cancer cell lines PC3, DU145, LNCaP, and C4-2B is more abundant compared to the normal non-malignant prostate epithelial cell lines RWPE-1, PZ-HP-7, and HPrEC (Supplementary Fig 1c).
2. In the PTEN-deficient prostate cancer cell line PC3, knockout of *ARID4B* efficiently inhibited tumor growth (Fig. 3b-e), suggesting the dependency on ARID4B by PTEN-deficient prostate cancer cells. Depletion of *ARID4B* in another PTEN-deficient prostate cancer cell line LNCaP also significantly suppresses cell growth (Supplementary Fig. 16). In the PTEN-intact prostate cancer cell line DU145, knockout of *ARID4B* only moderately suppressed tumor growth (Supplementary Fig. 7b-d).
3. Knockdown of *ARID4B* in human prostate cancer cell lines PC3, DU145, and LNCaP reduced expression or phosphorylation of the core regulators and downstream effectors of the PI3K-AKT pathway analyzed by western blot (Supplementary Fig. 2a-c).
4. We generated the PTEN-inducible PC3 cell line to test the effects of PTEN re-expression on ARID4B expression and PIK3CA-AKT signaling. Our results showed that upon PTEN re-expression, expression of ARID4B, PIK3CA, and PIK3R2 was reduced, resulting in the inhibition of the PI3K-AKT signaling (Fig. 5e and Supplementary Fig. 11a).
5. We showed that depletion of ARID4B inhibited colony formation in PC3 and DU145 cells (Supplementary Fig. 15a, b). Furthermore, re-expression of ARID4B in the *ARID4BKO* PC3 and *ARID4BKO* DU145 cells increased colony formation (Supplementary Fig. 15c).

Q4) The gene expression changes after knockdown of ARID4B in the PC3 cell line appear modest with no supporting Western blot data (Figure 1A-B).

A4: In the revised manuscript, we showed that knockdown of *ARID4B* in human prostate cancer cell lines PC3, DU145, and LNCaP reduced expression or phosphorylation of the core regulators and downstream effectors of the PI3K-AKT pathway by western blot analyses (Supplementary Fig. 2a-c).

Q5) The authors use transient expression of luciferase reporters to measure the effect of ARID4B on gene transcription. However, the authors propose that ARID4B works as a chromatin remodeler. Therefore, the luciferase experiments do not provide an appropriate model for their studies because plasmid DNA does not under chromatinization 48 hours after transfection.

A5: Although the lack of chromatinization in the transiently transfected plasmids has been reported (ref. 16), other reports demonstrated the association of nucleosomes on plasmid DNA after transfection into mammalian cells (refs. 17-19), suggesting that plasmid DNA is packaged into chromatin-like structures. In addition, some studies showed that chromatin remodeling proteins are able to induce effects on the gene promoters in plasmids to regulate expression of the luciferase reporter (refs. 20, 21). Similarly, our results from luciferase reporter gene assays suggest that ARID4B activates the *PIK3CA* and *PIK3R2* promoters (Fig. 1c).

References

16. Snowden AW, *et al.* Gene-specific targeting of H3K9 methylation is sufficient for initiating repression in vivo. *Curr. Biol.* 12, 2159–2166 (2002).
17. Reeves R, *et al.* Minichromosome assembly of non-integrated plasmid DNA transfected into mammalian cells. *Nucleic Acids Res.* 13, 3599–3615 (1985).
18. Jeong S. and Stein A. Micrococcal nuclease digestion of nuclei reveals extended nucleosome ladders having anomalous DNA lengths for chromatin assembled on non-replicating plasmids in transfected cells. *Nucleic Acids Res.* 22, 370–375 (1994).
19. Mladenova V, Mladenov E., and Russev, G. Organization of plasmid DNA into nucleosome-like structures after transient transfection in eukaryotic cells. *Biotechnol. Biotech. Eq.* 23, 1044–1047 (2009).
20. Li H, *et al.* The histone methyltransferase SETDB1 and the DNA methyltransferase DNMT3A interact directly and localize to promoters silenced in cancer cells. *J. Biol. Chem.* 281, 19489–19500 (2006).
21. Huang ZQ, *et al.* A role for cofactor-cofactor and cofactor-histone interactions in targeting p300, SWI/SNF and mediator for transcription. *EMBO J.* 22, 2146–2155 (2003).

Q6-1) The ChIP-seq data in Figures 2a&b and SuppFigure3A are problematic. The similarity between the signals, especially in the heat map, between the ARID4B and Input DNA suggest that the authors may have sheared the DNA too much. If so, they would tend to find their signals around TSS. It appears that they used Active Motif to carry out the bioinformatics analysis. They should

enlist the expertise of an academic bioinformatician to re-analyze these data. The ChIP-pcr data are more convincing of recruitment of ARID4B to the promoters of PIK3CA and PIK3R2. However, showing that ARID4B is absent from the MYOD1 promoter is not sufficient to establish the specificity of ARID4B binding throughout the genome. Perhaps ARID4B is present at all actively transcribed genes. Did the authors integrate the RNA-seq data to the ChIP-seq data to address this association?

A6-1: In the ChIP-seq data, input results often show small peaks at TSSs, because open promoter regions are more sensitive to sonication, while inactive chromatin can be very difficult to shear (refs. 22, 23). We used the peak finder tool MACS (v1.4.2) to look for regions that are more enriched in the ARID4B ChIP sample over Input, so the background peaks in this region can be taken into account, which is the reason we included an input as the ChIP-seq control. We now included the promoter average plot in the revised manuscript, which clearly showed that the promoter enrichment peak of ARID4B ChIP is markedly stronger compared to the Input peak (Supplementary Fig. 4a).

SFig. 4a

References

22. Landt SG, *et al.* ChIP-seq guidelines and practices of the ENCODE and modENCODE consortia. *Genome Research* 22, 1813-1831 (2012).
23. Auerbach RK, *et al.* Mapping accessible chromatin regions using Sono-Seq. *PNAS* 106, 4926–14931 (2009).

Q6-2) The ChIP-seq data in Figures 2a&b and SuppFigure3A are problematic. The similarity between the signals, especially in the heat map, between the ARID4B and Input DNA suggest that the authors may have sheared the DNA too much. If so, they would tend to find their signals around TSS. It appears that they used Active Motif to carry out the bioinformatics analysis. They should enlist the expertise of an academic bioinformatician to re-analyze these data. **The ChIP-pcr data are more convincing of recruitment of ARID4B to the promoters of PIK3CA and PIK3R2. However, showing that ARID4B is absent from the MYOD1 promoter is not sufficient to establish the specificity of ARID4B binding throughout the genome. Perhaps ARID4B is present at all actively transcribed genes. Did the authors integrate the RNA-seq data to the ChIP-seq data to address this association?**

A6-2: In the revised manuscript, we showed that ARID4B was not recruited to the promoters of *GAPDH* and *ARID4B* itself by ChIP-qPCR analysis (Supplementary Fig. 4b). In PC3 cells, *GAPDH* and *ARID4B* are actively expressed (Fig. 1b), and yet the ARID4B protein was not recruited to their promoters, indicating the recruitment of ARID4B to the *PIK3CA* and *PIK3R2* promoters is specific.

In addition, we have integrated the ChIP-seq data with the RNA-seq data (Supplementary Table 3). In all, 3,173 genes were identified by RNA-Seq (P -value < 0.05 , $|\text{Log}_2\text{FC}| \geq 0.59$) (Supplementary Table 1), and 11,675 genes were identified by ChIP-Seq analyses (Supplementary Table 2). Among these genes, 1,554 genes were identified by both RNA-Seq and ChIP-Seq analyses (Supplementary Table 3).

Together, our ChIP-Seq and ChIP-qPCR analyses showed that ARID4B was specifically recruited to the *PIK3CA* and *PIK3R2* promoters, and did not bind to all actively transcribed genes.

Q7-1) The authors observe a negative correlation between the presence of ARID4B and histone H1 at 3 promoters and suggest that this provides the mechanism for how ARID4B activates expression of *PIK3CA* and *PIK3R2*. Their limited data provide a tenuous connection. To support this notion, they need to compare H1 ChIP-seq to ARID4B ChIP-seq in the presence or absence of ARID4B expression. It was also not clear why they used H3K4me3 as a control for these experiments. Considering that ARID4B is a component of a HDAC complex, wouldn't it have made more sense to examine H3 and H4 acetylation?

A7-1: We used ChIP-seq to map global binding sites for ARID4B. Then, ChIP-qPCR was used to further confirm the ARID4B binding to the *PIK3CA* and *PIK3R2* promoters.

To investigate the mechanism by which ARID4B activates the *PIK3CA* and *PIK3R2* promoters, we specifically focused on the *PIK3CA* and *PIK3R2* promoters and analyzed whether ARID4B has an impact on the binding of histone H1 to these promoters. We did not intend to investigate the impact of ARID4B on the global H1 binding sites. Since ChIP-seq is for mapping global binding sites of DNA binding proteins and ChIP-qPCR examines specific DNA binding sites by proteins of interest, we performed ChIP-qPCR, instead of H1 ChIP-seq, to investigate the H1 binding specific on the *PIK3CA* and *PIK3R2* promoters in the presence or absence of ARID4B. ChIP-qPCR analyses showed that ablation of ARID4B results in increased binding of histone H1 to the *PIK3CA* and *PIK3R2* promoters (Fig. 2d).

Q7-2) The authors observe a negative correlation between the presence of ARID4B and histone H1 at 3 promoters and suggest that this provides the mechanism for how ARID4B activates expression of *PIK3CA* and *PIK3R2*. Their limited data provide a tenuous connection. To support this notion, they need to compare H1 ChIP-seq to ARID4B ChIP-seq in the presence or absence of ARID4B expression. It was also not clear why they used H3K4me3 as a control for these experiments. Considering that ARID4B is a component of a HDAC complex, wouldn't it have made more sense to examine H3 and H4 acetylation?

A7-2: In this revised manuscript, we showed that ablation of ARID4B in PC3 cells did not affect the levels of H4 acetylation on the promoters of *PIK3CA* and *PIK3R2* (Supplementary Fig. 5a), suggesting that ARID4B does not mediate the HDAC1 activity on these two promoters.

In addition, depletion of HDAC1 by two different short interfering RNAs SiHDAC1-1 and SiHDAC1-2 resulted in increased expression of PIK3CA and PIK3R2 in prostate cancer cell lines PC3, DU145, and LNCaP (Supplementary Fig. 14a-c). These results suggest that HDAC1 suppressed expression of PIK3CA and PIK3R2. On the other hand, we showed that ARID4B activated expression of PIK3CA and PIK3R2 in prostate cancer cell lines PC3, DU145, and LNCaP (Fig 1a, b, and Supplementary Fig. 2a-c). Our results indicate the opposing effects of ARID4B and HDAC1 on regulation of *PIK3CA* and *PIK3R2* expression.

Together, our results suggest that ARID4B and HDAC1 are likely to function independently of each other to regulate *PIK3CA* and *PIK3R2* in prostate cancer cells. Therefore, the SIN3A/HDAC1 repressor complex does not contribute to activation of *PIK3CA* and *PIK3R2* by ARID4B.

Q8-1) The authors do not provide any statistical analyses for the primary tumor data in Figure 3A. Furthermore, the rest of the data in this figure demonstrate that loss of ARID4D almost completely abrogates growth of the PC3 cell line. Therefore, the fact that the cells don't grow in tumorspheres or in mice doesn't offer much insight into its mechanism of action.

A8-1: In this revised manuscript, we included the statistical *P* values in analysis of mutual exclusivity between *ARID4B* and *PTEN* deletions from the prostate cancer genome datasets in Supplementary Fig. 6 as follows.

Cohort	Sample size	ARID4B alteration rate (n)	PTEN alteration rate (n)	Tendency	P value
Armenia et al. Nat Genetics 2018	1,013	3% (30)	16 % (162)	Mutual exclusivity	0.500
TCGA, Cell 2015	333	4 % (13)	17 % (56)	Mutual exclusivity	0.599
Robinson et al. Cell 2015	150	2.7 % (4)	40 % (60)	Mutual exclusivity	0.649
Kumar et al. Nat Med 2016	54	9 % (4)	44 % (23)	Mutual exclusivity	0.110
Taylor et al. Cancer Cell 2010	103	1 % (1)	14 % (14)	Mutual exclusivity	0.854

Supplementary Fig. 6 The tendency of mutual exclusivity between *ARID4B* and *PTEN* deletions. Analysis of five prostate cancer genome datasets using the cBioportal data source (<http://www.cbioportal.org/>) suggests a tendency of mutual exclusivity between *ARID4B* and *PTEN* deletions in prostate cancer genome.

Although statistical analysis of the *ARID4B-PTEN* mutual exclusivity in prostate cancer datasets didn't show the significance ($P > 0.05$) (Supplementary Fig. 6), a tendency of mutual exclusivity between *ARID4B* and *PTEN* deletions in prostate cancer genome was observed in all five datasets.

In most cohorts, *ARID4B* gene alteration rate is less than 5%. It is possible that the extremely few cases of *ARID4B* gene alterations render it mathematically difficult to obtain a significant *P* value. Similar to our findings, a tendency of mutual exclusivity between *CHD1* and *PTEN* deletions in prostate and breast cancer was suggested without reaching statistical significance (ref. 24). It also has been reported that even though the mutual exclusivity between *ARID1B* and *KRAS* is not significant in any individual cancer type, the pair show statistical significance for mutual exclusivity when multiple cancer types were combined to increase the sample number and to reach the statistical power (ref. 25).

Importantly, the tendency of ARID4B-PTEN mutual exclusivity led us to further evaluate the intricate relations between these two genes. The extensive *in vitro* and *in vivo* data, including those from depletion of ARID4B in the PTEN-deficient prostate cancer cell line PC3 and the prostate-specific *ARID4B* and *PTEN* double knockout mouse model, demonstrated the dependency on ARID4B by PTEN-deficient prostate cancer cells.

In the Discussion section of this revised manuscript, we included the following paragraph.

“Analysis of five prostate cancer genome datasets using the cBioportal data source (<http://www.cbioportal.org/>) suggests a tendency of mutual exclusivity between *ARID4B* and *PTEN* deletions in prostate cancer genome, although it does not reach the statistical significance ($P > 0.05$) (Supplementary Fig. 6). In most cohorts, *ARID4B* gene alteration rate is less than 5%. It is possible that the extremely few cases of *ARID4B* gene alterations render it mathematically difficult to obtain a significant *P* value. Similar to our findings, previous report by others also suggests a tendency of mutual exclusivity between *CHD1* and *PTEN* deletions in prostate and breast cancer without reaching statistical significance (ref. 24). It also has been reported that even the mutual exclusivity between *ARID1B* and *KRAS* is not significant in any individual cancer type, but the pair show statistical significance for mutual exclusivity when multiple cancer types were combined to increase the sample number and to reach the statistical power (ref. 25). Importantly, the tendency of ARID4B-PTEN mutual exclusivity led us to further evaluate the intricate relations between these two genes. The extensive *in vitro* and *in vivo* data, including those from depletion of ARID4B in the PTEN-deficient prostate cancer cell line PC3 and the prostate-specific *ARID4B* and *PTEN* double knockout mouse model, demonstrated the dependency on ARID4B by PTEN-deficient prostate cancer cells.”

References

24. Zhao D *et al.* Synthetic essentiality of chromatin remodeling factor CHD1 in PTEN deficient cancer. *Nature* 542, 484–488 (2017)
25. Kim Y.A. *et al.* MEMCover: integrated analysis of mutual exclusivity and functional network reveals dysregulated pathways across multiple cancer types. *Bioinformatics* 31, i284–i292 (2015)

Q8-2) The authors do not provide any statistical analyses for the primary tumor data in Figure 3A. **Furthermore, the rest of the data in this figure demonstrate that loss of ARID4D almost completely abrogates growth of the PC3 cell line. Therefore, the fact that the cells don't grow in tumorspheres or in mice doesn't offer much insight into its mechanism of action.**

A8-2: To provide a collective understanding on how loss of ARID4B affects behavior of tumor cells, we investigated cancer cell growth in different culture conditions. Therefore, the control and *ARID4B*KO PC3 cells were analyzed by being grown in the culture plates, tumorsphere formation, and the xenograft mouse model.

In culture plates, it provides cancer cells a surface where they are grown in monolayer. In this culture condition, we found that knockout of *ARID4B* in PC3 cells efficiently constrained cell growth (Fig. 3b).

However, prostate tumors grow in a three-dimensional (3D) spatial conformation, resulting in a heterogeneous exposure to oxygen and nutrients. To mimic the 3D spatial conformation, we examined tumorsphere formation of the control and *ARID4B*KO PC3 cells that were grown in ultra-low attachment culture plates (Fig. 3c).

Even so, cell culture is the process by which cancer cells are grown outside their natural environment. Therefore, we employed the xenograft mouse model to create an environment that allows for the natural tumor growth of the control and *ARID4B*KO PC3 cells in mice (Fig. 3d).

Q9) The authors present no comparison of the 3 gene signature as a predictor for prognosis of prostate cancer recurrence to other available prognostic signatures.

A9: In the Discussion section of the revised manuscript, we compared the *PTEN/ARID4B/PIK3CA* three gene signature with the *PTEN/SMAD4/CCND1/SPP1* four gene signature (ref. 26) as follows.

“Similar to the *PTEN/ARID4B/PIK3CA* three gene signature, it has been reported that *PTEN* together with *SMAD4*, *CCND1*, and *SPP1* form a four gene signature that is prognostic of recurrence and lethal metastasis in human prostate cancer (ref. 26). *SMAD4* is a key regulator of the TGF β pathway and is a suppressor of prostate tumor progression, whereas cyclin D1 and *SPP1* act as mediators of the prostate cancer processes. The inactivation of *PTEN* and *SMAD4* as well as activation of cyclin D1 and *SPP1* promote prostate cancer. Yet, combination of *PTEN/SMAD4* do not by themselves predict risk for prostate cancer recurrence. However, when combined with Cyclin D1 and *SPP1*, the prognostic value of the *PTEN/SMAD4/CCND1/SPP1* four gene signature were enhanced, providing significant prediction for prostate cancer recurrence risk. This four-gene model distinguished the high-risk patient group from the low-risk group for the prostate cancer recurrence by Kaplan–Meier analysis [hazard ratio (HR) = 2.6, $P = 0.012$]. Similarly, we showed the *PTEN/ARID4B/PIK3CA* three-gene combinatorial signature has the most predictive power for tumor recurrence over that of individual gene alone or any two genes combined. This *PTEN/ARID4B/PIK3CA* signature robustly stratified patients into high- and low-risk groups for tumor recurrence (HR = 4.58, $P = 0.006$ in the GSE40272 dataset, and HR = 3.23, $P = 0.002$ in

the GSE21032 dataset). In addition, the *PTEN/SMAD4/CCND1/SPP1* four gene signature involves in the PTEN and TGF β pathways that converge to regulate the prostate cancer progression (ref. 26). On the other hand, we identified the PTEN-ARID4B-PI3K regulatory axis that illustrates the molecular mechanism underlying the predictive power of the *PTEN/ARID4B/PIK3CA* signature for tumor recurrence in prostate cancer patients.”

Reference

26. Ding Z *et al.* SMAD4-dependent barrier constrains prostate cancer growth and metastatic progression, *Nature* 470, 269–276 (2011)

Q10-1) The authors do not present key controls such as- does reexpression of ARID4B rescue the growth effects observed in the knockout cell line and does the absence of ARID4B prevent increased PIK3CA and PIK3R2 expression upon PTEN reexpression?

A10-1: In this revised manuscript, we showed that depletion of ARID4B inhibited colony formation in PC3 and DU145 cells (Supplementary Fig. 15a, b). Furthermore, re-expression of ARID4B in the *ARID4BKO* PC3 and *ARID4BKO* DU145 cells increased colony formation (Supplementary Fig. 15c).

Q10-2) The authors do not present key controls such as- does reexpression of ARID4B rescue the growth effects observed in the knockout cell line and **does the absence of ARID4B prevent increased PIK3CA and PIK3R2 expression upon PTEN reexpression?**

A10-2: In this revised manuscript, we showed that re-expression of PTEN in the PTEN-deficient prostate cancer cell PC3 resulted in decreased expression of ARID4B, accompanied by down-regulation of PIK3CA and PIK3R2 and inactivation of the PI3K-AKT signal pathway (Fig. 5e and Supplementary Fig. 11a).

Q11) The Discussion is far too long, mainly due to the authors’ repetition of the results. The authors should focus on interpreting the basic science and translational implications of their results.

A11: In this revised manuscript, we revised the Discussion section and discussed the basic science and translational implications.

Reviewers' comments:

Reviewer #1 (Remarks to the Author):

Oncogenic PI3K signalling is frequently activated in prostate cancer, and is a major driver of tumour growth and resistance to the current standard of care. This study investigates the functional importance of the chromatin remodeling factor ARID4B in human prostate cancer cell lines and transgenic mouse models of prostate cancer. The authors identify ARID4B as a transcriptional activator of PIK3CA and PIK3R2 that encode catalytic and regulatory PI3K subunits respectively, presenting a new potential branch of the complex PI3K signalling pathway. The authors also report that ARID4B deletion prevents carcinoma progression in a GEMM of prostate cancer deficient for the tumour suppressor PTEN that causes oncogenic PI3K signalling, and that ARID4B CNVs/mutations occur in prostate cancer patients, while mRNA expression can correlate with worse outcome.

However, the functional importance of PIK3CA and PIK3R2 upregulation and the clinical relevance of the findings remains unclear and the current manuscript requires revision based on the following points:

Specific points:

1. The authors show ARID4B upregulates PIK3CA and PIK3R2 transcription. Can upregulated PIK3CA/PIK3R2 mRNA contribute to prostate cancer progression?
2. Currently, the data in Fig. 5h, Supl Figs. 12/13 is over-interpreted in the abstract, results and discussion, as the predictive value of ARID4B and the 3-gene signature needs to be verified in larger patient cohorts to support this claim. Do larger, comprehensive prostate cancer patient datasets with mRNA data also show a similar correlation (e.g. TCGA provisional, Taylor 2010, Robinson 2015)? Can ARID4B CNA status/mutations predict for worse survival? or just ARID4B mRNA? To validate findings, the authors should analyze ARID4B expression in relation to PTEN using a prostate cancer tissue microarray with clinical follow-up data.
3. The methodology of how the 3-gene signature was generated is unclear and needs to be included. For instance, how were the high-risk and low-risk groups defined?
4. The data presented suggest that ARID4B is not simply mediated by PTEN. Although xenograft experiments revealed the ARID4B KD rescue effect is maintained in the PTEN-null PC3 cells, a significant response was also observed in DU145 cells that have not lost PTEN. What is the mechanism whereby PTEN modulates ARID4B? How do PTEN-intact prostate cancer cells respond to ARID4B loss, and why does resistant growth arise?

Furthermore, Fig. 5j indicates PTEN directly inhibits ARID4B, yet no evidence is presented to support this.

5. Since the authors have previously reported ARID4B is an Androgen Receptor co-activator (ref 39), the authors should determine if androgen signalling is deregulated in the Arid4b;Pten-deficient GEMM of prostate cancer relative to the Pten-deficient model by qRT-PCR for AR regulated genes/IHC for AR.

Minor points:

- a) the first line of the abstract and second line of the introduction; "PTEN is the most frequently mutated gene in prostate cancer" is not true. While PTEN is commonly mutated in prostate cancer, there are several recent publications indicating other genetic aberrations occur with a higher frequency such as TP53, SPOP and AR (e.g. Figure 1: Armenia 2018 Nat Genetics).
- b) Second sentence of the results section "Analysis showed that amplification of ARID4B is

detected in some human prostate cancer cases...." needs to be more specific in terms of the number of patients.

c) N-values and % altered statistics performed categorized by mRNA down-/up-regulated should be added to Suppl Fig. 12/13 to enable further interpretation of the data presented.

d) In terms of article organization, data from the transgenic mouse model deficient for Pten and Aridb4 is shown in figure 1, yet the GEMMs are not characterised until Figure 4. The authors should move Fig. 1f to Fig. 4. Patient data should also be discussed together, thus patient data in figure 3, should be merged into Fig 1.

e) Throughout the manuscript, the authors incorrectly refer to the genes PIK3CA and PIK3R2 as proteins. For instance, gene nomenclature is incorrectly used to label protein bands in Western blot images and schematic pathway diagrams and needs to be corrected. For instance, the PIK3CA gene encodes the PI3K catalytic subunit p110alpha.

f) Lines 177-180 are scientifically incorrect and should be amended. Truncating mutations in PTEN in the Armenia dataset for example do occur in prostate cancer specimens that carry ARID4B deep deletion, indicating that ARID4B is not consistently retained in patients that have lost PTEN.

Reviewer #2 (Remarks to the Author):

The studies in this revised manuscript support a model where ARID4B overexpression drives prostate cancer development by directly increasing expression of PIK3CA and PIK3R2 expression. The authors demonstrate that decreased expression of ARID4B in PTEN-deficient prostate cancer cell lines and in a genetically-engineered mouse model of prostate cancer significantly inhibits growth in culture and in vivo. They further demonstrate that ARID4B activates gene expression through competition with H1 histones. In addition, they show that high ARID4B expression in primary prostate cancers shows high predictive power for poorer overall survival, either alone or in combination with other clinical markers. Importantly, the authors provide significant new data to address the concerns of the previous reviewers. Thus, the current version identifies a potentially exciting new regulatory mechanism for PTEN-PI signaling. Once the authors have addressed the remaining issues below, this manuscript should prove of interest to a broad range of basic and translational researchers.

Major Criticisms:

1) If ARID4B activates the PI3K pathway, why do the majority of prostate cancers in the next generation sequencing studies show deletion of ARID4B? This appears inconsistent with the authors' model.

2) The growth inhibition in vivo shown in Figure S7C would be considered potent in most other reports. This result appears relevant to concern in comment#1. Why do many PTEN-positive prostate cancers display deletion of ARID4B?

3) The authors should discuss why the predictive power of ARID4B alone (Figure S1d) appears more robust than the 3 gene signature in Figure 5i.

4) The origin, growth conditions and verification of the cell lines used in Supplementary Studies-LNCaP, C4-2B, RWPE-1, PZ-HP-7 and HPrEC- were not included in the manuscript.

5) Have the authors validated the gene expression studies by IHC of primary human prostate tumors?

Minor Criticisms:

1) The manuscript contains numerous grammatical errors that the authors should have addressed before submitting a revised manuscript.

Subject: Decision on manuscript NCOMMS-18-28842A.

Reviewers' comments:

Reviewer #1 (Remarks to the Author):

Oncogenic PI3K signaling is frequently activated in prostate cancer, and is a major driver of tumour growth and resistance to the current standard of care. This study investigates the functional importance of the chromatin remodeling factor ARID4B in human prostate cancer cell lines and transgenic mouse models of prostate cancer. The authors identify ARID4B as a transcriptional activator of PIK3CA and PIK3R2 that encode catalytic and regulatory PI3K subunits respectively, presenting a new potential branch of the complex PI3K signaling pathway. The authors also report that ARID4B deletion prevents carcinoma progression in a GEMM of prostate cancer deficient for the tumour suppressor PTEN that causes oncogenic PI3K signalling, and that ARID4B CNVs/mutations occur in prostate cancer patients, while mRNA expression can correlate with worse outcome.

However, the functional importance of PIK3CA and PIK3R2 upregulation and the clinical relevance of the findings remains unclear and the current manuscript requires revision based on the following points:

Specific points:

Q1. The authors show ARID4B upregulates PIK3CA and PIK3R2 transcription. Can upregulated PIK3CA/PIK3R2 mRNA contribute to prostate cancer progression?

A1. We wish to clarify that our results on regulation of *PIK3CA* and *PIK3R2* by ARID4B are obtained from the “loss” of ARID4B function experiments. In this manuscript, we showed that knockout of *ARID4B/Arid4b* in PTEN-deficient prostate cancer PC3 cells and the *Pten*^{PC-/-} mice resulted in down-regulation of *PIK3CA/Pik3ca* and *PIK3R2/Pik3r2* (Fig. 1a, b, e, and Fig. 4e) and inhibition of the PI3K/AKT signaling pathway (Fig. 1e and Fig. 4e), leading to suppression of cell proliferation, tumor formation, and cancer progression (Figs. 3 and 4). These results suggest that ARID4B is crucial for *PIK3CA* and *PIK3R2* expression, which is required for activation of the PI3K/AKT pathway. It is well known that activation of the PI3K/AKT signaling is associated with prostate cancer progression, particularly, in PTEN-deficient prostate cancer (Martini et al., 2014).

The reporter gene assays in Fig. 1c are to support that ARID4B regulates the activity of the *PIK3CA* and *PIK3R2* promoters. This result is not intended to suggest that ARID4B up-regulates *PIK3CA* and *PIK3R2* to promote prostate cancer. We apologized for this confusion.

References

1. Martini, M., De Santis, M. C., Braccini, L., Gulluni, F., and Hirsch, E. (2014). PI3K/AKT signaling pathway and cancer: an updated review. *Ann Med* 46, 372-383.

Q2-1. Currently, the data in Fig. 5h, Supl Figs. 12/13 is over-interpreted in the abstract, results and

discussion, as the predictive value of ARID4B and the 3-gene signature needs to be verified in larger patient cohorts to support this claim. Do larger, comprehensive prostate cancer patient datasets with mRNA data also show a similar correlation (e.g. TCGA provisional, Taylor 2010, Robinson 2015)?

A2-1. In the Results section of this revised manuscript, we analyzed 3 datasets, including the TCGA provisional (n = 492) (Supplementary Fig. 13c, e), GSE21032 (n = 140) (Taylor et al., 2010) (Supplementary Fig. 13b, d), and GSE40272 (n = 89) (Gulzar et al., 2013) (Fig. 6d, e) (Fig. 5h in previous version). The results from all three datasets support our findings and showed the enhanced prognostic value of *PTEN*, *ARID4B*, and *PIK3CA* in combination for tumor recurrence in prostate cancer patients.

The Robinson 2015 dataset does not contain clinical information (Robinson et al., 2015).

References

1. Gulzar, Z. G., McKenney, J. K., and Brooks, J. D. (2013). Increased expression of NuSAP in recurrent prostate cancer is mediated by E2F1. *Oncogene* 32, 70-77.
2. Taylor, B. S., Schultz, N., Hieronymus, H., Gopalan, A., Xiao, Y., Carver, B. S., Arora, V. K., Kaushik, P., Cerami, E., Reva, B., *et al.* (2010). Integrative genomic profiling of human prostate cancer. *Cancer Cell* 18, 11-22.
3. Robinson, D., Van Allen, E. M., Wu, Y. M., Schultz, N., Lonigro, R. J., Mosquera, J. M., Montgomery, B., Taplin, M. E., Pritchard, C. C., Attard, G., *et al.* (2015). Integrative clinical genomics of advanced prostate cancer. *Cell* 161, 1215-1228.

Q2-2. Can ARID4B CNA status/mutations predict for worse survival? or just ARID4B mRNA?

A2-2. The sample numbers of ARID4B CNA/mutations in prostate cancer genome datasets with available clinical information (Fig. 3a and Supplementary Fig. 1a) are too low to reach the statistical power for predicting prostate cancer recurrence or survival in patients. For examples, there are 18 prostate cancer patients carrying ARID4B CNA/mutations in TCGA provisional dataset, 4 prostate cancer patients carrying ARID4B CNA/mutations in the Grasso dataset (Grasso et al., 2012), and 1 prostate cancer patient carrying ARID4B CNA/mutations in the Taylor dataset (Taylor et al., 2010).

In this manuscript, results are based on the levels of ARID4B mRNA, which showed that expression of *ARID4B* in prostate tumors is negatively associated with recurrence-free survival in prostate cancer patients.

References

1. Grasso, C. S., Wu, Y. M., Robinson, D. R., Cao, X., Dhanasekaran, S. M., Khan, A. P., Quist, M. J., Jing, X., Lonigro, R. J., Brenner, J. C., *et al.* (2012). The mutational landscape of lethal castration-resistant prostate cancer. *Nature* 487, 239-243.
2. Taylor BS, *et al.* Integrative genomic profiling of human prostate cancer. *Cancer Cell* 18, 11-22 (2010).

Q2-3. To validate findings, the authors should analyze ARID4B expression in relation to PTEN using a prostate cancer tissue microarray with clinical follow-up data.

A2-3. As requested, we performed immunohistochemical staining on prostate cancer tissue microarrays (TMAs) to investigate the clinical relevance of the newly identified PTEN-ARID4B-PI3K axis in human prostate cancer.

The results support our findings and suggest a negative correlation between ARID4B and PTEN expression levels in human prostate adenocarcinoma (Fig. 6b, c and Supplementary Fig. 13a).

In the Results section, we added the following paragraph.

“To investigate the clinical relevance of the newly identified PTEN-ARID4B-PI3K axis in human prostate cancer, we performed immunohistochemical staining with antibodies against ARID4B, PTEN, and p110 α on tissue microarrays (TMAs) consisting of 118 prostate adenocarcinoma samples from 85 patients and 73 prostate hyperplasia samples from 36 patients. Results revealed that ARID4B levels were higher in samples with prostate adenocarcinoma compared to prostate hyperplasia (Fig. 6a, c, left and Supplementary Fig. 13a, left), consistent with results from OncoPrint database analysis showing increased expression of *ARID4B* in human prostate carcinoma compared to normal prostate glands (Supplementary Fig. 1b). In addition, levels of PTEN were diminished in prostate adenocarcinoma (Fig. 6a, c, middle and Supplementary Fig. 13a, middle), whereas the levels of p110 α were elevated (Fig. 6a, c, right and Supplementary Fig. 13a, right). Furthermore, expression of ARID4B and PTEN were negatively correlated in human prostate adenocarcinoma (Fig. 6b, c and Supplementary Fig. 13a), whereas ARID4B and p110 α levels were positively correlated in human prostate adenocarcinoma (Fig. 6b, c and Supplementary Fig. 13a).”

Q3. The methodology of how the 3-gene signature was generated is unclear and needs to be included. For instance, how were the high-risk and low-risk groups defined?

A3. As requested, the methodology is added to the Methods section as the following paragraph.

“Calculation of prognostic index for the *PTEN/ARID4B/PIK3CA* three-gene signature

In Fig. 6d, e and Supplementary Fig. 13b-e, we demonstrated the predictive power of the *PTEN/ARID4B/PIK3CA* three-gene signature for tumor recurrence in three datasets: TCGA provisional dataset, GSE21032, and GSE40272. The prognostic index (PI) was used to define the high and low risk scores, and the two risk groups (high and low) were generated by splitting patients with the median PI. The PI, as the linear component of the Cox model, was calculated as

$PI = b_1x_1 + b_2x_2 + \dots + b_px_p$, where x is the gene expression value and the b is a risk coefficient obtained from the Cox fitting. The fitting was performed in R (<http://cran.r-project.org>) using the survival package.

The multivariate Cox proportional hazards model of the *ARID4B/PTEN/PIK3CA* signature combined expression levels of these three genes with other clinical and pathological variables into an integrated risk score model to predict time to recurrence in prostate cancer patients from the TCGA provisional dataset using statistical software SPSS19.0 and from two datasets GSE21032² and GSE40272²⁸ using the SurvExpress web tool⁴⁷ (<http://bioinformatica.mty.itesm.mx:8080/Biomatec/SurvivaX.jsp>). Briefly, the PI was used to define and generate high and low risk groups with classical multivariate Cox model, with higher PI representing higher risk. Groups at high and low risk for recurrence were generated by splitting values at the median PI. In addition, Kaplan-Meier survival curves were used to show that the *ARID4B/PIK3CA/PTEN* gene could distinguish between the two groups for the prostate cancer recurrence. Patients in high-risk groups had a lower survival rate than those in the low-risk group in the TCGA provisional, GSE21032 ($P = 0.002$), and GSE40272 ($P = 0.006$) datasets.”

References

1. Aguirre-Gamboa, R., Gomez-Rueda, H., Martinez-Ledesma, E., Martinez-Torteya, A., Chacolla-Huaringa, R., Rodriguez-Barrientos, A., Tamez-Pena, J. G., and Trevino, V. (2013). SurvExpress: an online biomarker validation tool and database for cancer gene expression data using survival analysis. *PLoS One* 8, e74250.

Q4-1. The data presented suggest that ARID4B is not simply mediated by PTEN. Although xenograft experiments revealed the ARID4B KD rescue effect is maintained in the PTEN-null PC3 cells, a significant response was also observed in DU145 cells that have not lost PTEN. What is the mechanism whereby PTEN modulates ARID4B? How do PTEN-intact prostate cancer cells respond to ARID4B loss, and why does resistant growth arise?

Furthermore, Fig. 5j indicates PTEN directly inhibits ARID4B, yet no evidence is presented to support this.

A4-1. In this revised manuscript, we included results to show that PTEN inhibited ARID4B via an AKT-ARID4B feedback loop. These results are presented in Fig. 5h-k, and the following paragraph is added to the Results section.

“PTEN is dominated by its lipid phosphatase activity that counters PI3K actions, and thus deficiency of PTEN activates the PI3K-AKT signaling pathway⁶. To demonstrate that deficiency of PTEN increases expression of ARID4B *via* activation of the PI3K-AKT pathway, we showed that knockdown of *AKT* in the PTEN-deficient prostate cell line PC3 decreased expression of ARID4B (Fig. 5h). Consistent with this result, inhibition of the AKT activity by adding an AKT inhibitor MK-2206 downregulated ARID4B in PC3 cells (Fig. 5i). Furthermore, the *ARID4B* promoter-driven luciferase reporter gene assay showed that a constitutively active form of AKT (CA-AKT) activated the promoter of *ARID4B* (Fig. 5j). These results indicate that AKT positively regulates expression of *ARID4B*. Together with the fact that ARID4B is required for activation of

the PI3K-AKT signaling pathway (Fig. 1), our results suggest an AKT-ARID4B positive feedback loop.”

In the Discussion section of the revised manuscript, we added the following paragraph.

“Our study uncovers a new PTEN-ARID4B-PI3K axis and a novel AKT-ARID4B positive feedback loop whereby PTEN antagonizes the PI3K-AKT signaling pathway. In normal prostate, PTEN is attributed to its lipid phosphatase activity that directly counters the PI3K action, while ARID4B serves as a transcriptional activator of the PI3K subunit genes *PIK3CA* and *PIK3R2*, and expression of *ARID4B* itself is regulated by the AKT-ARID4B positive feedback loop (Fig. 5k, left). Deficiency of PTEN results in activation of the PI3K-AKT pathway and the AKT-ARID4B positive feedback loop, leading to prostate tumorigenesis (Fig. 5k, middle). In contrast, ablation of ARID4B inhibits the PI3K-AKT pathway, and compromises cancer progression in prostate harboring PTEN deficiency (Fig. 5k, right).”

Q4-2. The data presented suggest that ARID4B is not simply mediated by PTEN. Although xenograft experiments revealed the ARID4B KD rescue effect is maintained in the PTEN-null PC3 cells, a significant response was also observed in DU145 cells that have not lost PTEN. What is the mechanism whereby PTEN modulates ARID4B? How do PTEN-intact prostate cancer cells respond to ARID4B loss, and why does resistant growth arise?

A4-2. In this revised manuscript, we included data showing that the AKT inhibitor MK-2206 effectively inhibited cell growth in PC3 cells, and moderately suppressed cell growth in DU145 cells. This result suggest that the PTEN-deficient PC3 cells are heavily dependent on activation of the PI3K-AKT pathway for tumor growth, whereas the PTEN-intact DU145 cells partially rely on the PI3K-AKT signaling (Supplementary Fig. 17a, b). Therefore, knockout of *ARID4B* that inactivated the PI3K-AKT signaling pathway (Supplementary Fig. 3a) can also suppress cell growth and tumor formation in the PTEN-intact DU145 cells (Supplementary Fig. 7b, c), but not to the extent as in PC3 cells (Fig. 3b, d).

In the Discussion section, we added the following paragraph.

“In the PC3 (PTEN-deficient) cells, knockout of *ARID4B* efficiently inhibited cancer cell growth and tumor formation in xenografts, suggesting that PTEN-deficient prostate cancer cells depend on ARID4B. In the DU145 (PTEN-intact) cells, knockout of *ARID4B* also suppressed cell growth and tumor formation in xenografts, but not as much as in PC3 cells. PTEN-deficient PC3 cells may be more dependent on activation of the PI3K-AKT pathway for tumor growth than the PTEN-intact DU145 cells. This would explain why the AKT inhibitor MK-2206 inhibited cell growth more effectively in PC3 cells than DU145 cells (Supplementary Fig. 17a, b).

In PTEN-deficient PC3 cells, ablation of PTEN activates the PI3K/AKT pathway, leading to tumorigenesis (Supplementary Fig. 17c, left). In *ARID4B*KO PC3 cells, knockout of *ARID4B* compromises activation of the PI3K/AKT pathway, thus efficiently inhibiting tumorigenesis elicited by PTEN deficiency (Supplementary Fig. 17c, right). In PTEN-intact DU145 cells, PTEN dephosphorylates PIP3 into PIP2, which opposes the PI3K/AKT signaling pathway (Supplementary Fig. 17d, left), suggesting tumorigenesis of DU145 cells is less dependent on the PI3K-AKT signaling than PC3 cells. Therefore, knockout of *ARID4B* that inactivates the PI3K/AKT pathway only moderately suppressed tumorigenesis of DU145 cells (Supplementary Fig. 17d, right).

Fig. S17

In addition, ARID4B may promote cancer cell activity through additional pathways unrelated to PTEN tumor suppressor function. . For example, our immunohistochemical analyses detected increased numbers of cells positive for cleaved caspase 3, an apoptosis marker, in prostates of the *Pten*^{PC-/-}*Arid4b*^{PC-/-} mice compared to the *Pten*^{PC-/-} mice, suggesting that ablation of ARID4B promoted apoptosis (Supplementary Fig. 18a, b). Consistent with this result, expression levels of apoptosis regulators *p53* and *Fas* were elevated in prostates of the *Pten*^{PC-/-}*Arid4b*^{PC-/-} prostate compared to the *Pten*^{PC-/-} mice (Supplementary Fig. 18c). For its oncogenic function, ARID4B might directly or indirectly inhibit the apoptosis pathway mediated by the p53 tumor suppressor.”

Q5. Since the authors have previously reported ARID4B is an Androgen Receptor co-activator (ref 39), the authors should determine if androgen signalling is deregulated in the *Arid4b*;Pten-deficient GEMM of prostate cancer relative to the Pten-deficient model by qRT-PCR for AR regulated genes/IHC for AR.

A5. In this manuscript, we focused on the novel PTEN-ARID4B-PI3K axis by which ARID4B is required for PTEN-deficient prostate cancer.

In PC3 cells, ARID4B is required for cell growth (Fig. 3b) and tumor formation (Fig. 3d). PC3 cells are androgen receptor (AR)-negative. Therefore, function of ARID4B on PC3 cells does not depend on the AR signaling, although ARID4B is an AR co-activator.

In genetically engineered mouse models, prostate-specific knockout of *Arid4b* in mice compromised cancer progression in prostates harboring PTEN deficiency. Consistent with the results from PC3 cells with ablation of ARID4B (Fig. 1e), the PI3K-AKT signaling was reduced the *Pten*^{PC-/-}*Arid4b*^{PC-/-} prostate (Fig. 4e), suggesting a mechanism by which ARID4B is required for executing the prostate cancer-promoting actions in the context of PTEN deficiency. We agree with this reviewer that AR signaling might also contribute to ARID4B function in the *Pten*^{PC-/-} mice, but it is out of the scope of current investigation/manuscript which focused on function of the novel PTEN-ARID4B-PI3K axis.

Minor points:

a) the first line of the abstract and second line of the introduction; “PTEN is the most frequently mutated gene in prostate cancer” is not true. While PTEN is commonly mutated in prostate cancer, there are several recent publications indicating other genetic aberrations occur with a higher frequency such as TP53, SPOP and AR (e.g. Figure 1: Armenia 2018 Nat Genetics).

Answer (a): In the revised manuscript, the first sentence of the Abstract was changed from “*PTEN* is the most frequently mutated gene in prostate cancer.” to “*PTEN* is frequently mutated in prostate cancer.”

The first sentence of the Introduction was changed from “Loss of the tumor suppressor gene *PTEN* is the most frequent genetic alteration reported in prostate cancers.” to “Loss of the tumor suppressor gene *PTEN* is a frequent genetic alteration in prostate cancers”.

b) Second sentence of the results section “Analysis showed that amplification of ARID4B is detected in some human prostate cancer cases....” needs to be more specific in terms of the number of patients.

Answer (b): In the revised manuscript, the second sentence of the Results section was changed from “Analysis showed that amplification of ARID4B is detected in some human prostate cancer cases,” to “Analysis showed that amplification of *ARID4B* is detected in up to 20% of human prostate cancer cases,”

c) N-values and % altered statistics performed categorized by mRNA down-/up-regulated should be added to Suppl Fig. 12/13 to enable further interpretation of the data presented.

Answer (c): In the revised manuscript, we added “the dataset GSE40272 (n = 89)” in Supplementary Fig. 12 and “dataset GSE21032 (n = 140)” in Supplementary Fig. 13b.

d) In terms of article organization, data from the transgenic mouse model deficient for *Pten* and *Arid4b* is shown in figure 1, yet the GEMMs are not characterised until Figure 4. The authors should move Fig. 1f to Fig. 4. Patient data should also be discussed together, thus patient data in figure 3, should be merged into Fig 1.

Answer (d): As suggested, we moved data of the transgenic mouse model deficient for *Pten* and *Arid4b* from Fig. 1f to Fig. 4e.

e) Throughout the manuscript, the authors incorrectly refer to the genes *PIK3CA* and *PIK3R2* as proteins. For instance, gene nomenclature is incorrectly used to label protein bands in Western blot images and schematic pathway diagrams and needs to be corrected. For instance, the *PIK3CA* gene encodes the PI3K catalytic subunit p110 α .

Answer (e): In this revised manuscript, we used p110 α and p85 β as proteins encoded by *PIK3CA* and *PIK3R2*, respectively.

f) Lines 177-180 are scientifically incorrect and should be amended. Truncating mutations in *PTEN* in the Armenia dataset for example do occur in prostate cancer specimens that carry *ARID4B* deep deletion, indicating that *ARID4B* is not consistently retained in patients that have lost *PTEN*.

Answer (f): In the revised manuscript, we changed the sentence “we identified that *ARID4B* is consistently retained in *PTEN*-deficient prostate carcinoma cohorts, although deletion of *ARID4B* occasionally occurs in prostate cancer (Fig. 3a).” to “we identified that *ARID4B* is consistently retained in prostate carcinoma cohorts with deep deletion of *PTEN*, although deletion of *ARID4B* occasionally occurs in prostate cancer (Fig. 3a).”

In Fig. 3a (refs. 6-10), mutations in *PTEN* and *ARID4B* (truncating, missense, or inframe) in the Armenia and all other datasets are labeled as “putative driver” or “unknown significance”. There is no scientific evidence to suggest that these mutations activate or inhibit the activity of *PTEN* or *ARID4B*. In addition, there is no data to suggest whether these mutations promote or suppress prostate cancer. Furthermore, no scientific evidence suggest whether these mutations are *PTEN*-deficient or *ARID4B*-deficient.

References

1. Armenia J, *et al.* The long tail of oncogenic drivers in prostate cancer. *Nat Genet* **50**, 645-651 (2018).

2. Cancer Genome Atlas Research N. The Molecular Taxonomy of Primary Prostate Cancer. *Cell* **163**, 1011-1025 (2015).
3. Robinson D, *et al.* Integrative clinical genomics of advanced prostate cancer. *Cell* **161**, 1215-1228 (2015).
4. Kumar A, *et al.* Substantial interindividual and limited intraindividual genomic diversity among tumors from men with metastatic prostate cancer. *Nat Med* **22**, 369-378 (2016).
5. Taylor BS, *et al.* Integrative genomic profiling of human prostate cancer. *Cancer Cell* **18**, 11-22 (2010).

Reviewer #2 (Remarks to the Author):

The studies in this revised manuscript support a model where ARID4B overexpression drives prostate cancer development by directly increasing expression of PIK3CA and PIK3R2 expression. The authors demonstrate that decreased expression of ARID4B in PTEN-deficient prostate cancer cell lines and in a genetically-engineered mouse model of prostate cancer significantly inhibits growth in culture and in vivo. They further demonstrate that ARID4B activates gene expression through competition with H1 histones. In addition, they show that high ARID4B expression in primary prostate cancers shows high predictive power for poorer overall survival, either alone or in combination with other clinical markers. Importantly, the authors provide significant new data to address the concerns of the previous reviewers. Thus, the current version identifies a potentially exciting new regulatory mechanism for PTEN-PI signaling. Once the authors have addressed the remaining issues below, this manuscript should prove of interest to a broad range of basic and translational researchers.

Major Criticisms:

Q1) If ARID4B activates the PI3K pathway, why do the majority of prostate cancers in the next generation sequencing studies show deletion of ARID4B? This appears inconsistent with the authors' model.

A1) In this manuscript, we showed that the majority of *ARID4B* alterations in prostate cancer genome datasets is **amplification of *ARID4B*** with up to 23% of human prostate cancer cases (Supplementary Fig. 1a, left): the Beltran 2016 dataset (n = 114) showed 23% of human prostate cancer cases with amplification of ARID4B, the Abida 2019 dataset (n = 444) showed ~4% of human prostate cancer cases with amplification of ARID4B, the Grasso 2012 dataset (n = 61) showed ~3% of human prostate cancer cases with amplification of ARID4B, the Kumar 2016 dataset (n = 176) showed ~3% of human prostate cancer cases with amplification of ARID4B, although the TCGA provisional dataset (n = 499) showed 3% of human prostate cancer cases with deletion of ARID4B.

Q2) The growth inhibition in vivo shown in Figure S7C would be considered potent in most other reports. This result appears relevant to concern in comment#1. Why do many PTEN-positive prostate cancers display deletion of ARID4B?

A2) In this revised manuscript, we included data showing that the AKT inhibitor MK-2206 effectively inhibited cell growth in PC3 cells, and only moderately suppressed cell growth in DU145 cells. This result suggest that the PTEN-deficient PC3 cells are heavily dependent on activation of the PI3K-AKT pathway for tumor growth, whereas the PTEN-intact DU145 cells only partially rely on the PI3K-AKT signaling (Supplementary Fig. 17a, b). Therefore, knockout of *ARID4B* that inactivated the PI3K-AKT signaling pathway (Supplementary Fig. 3a) can also suppress cell growth and tumor formation in the PTEN-intact DU145 cells (Supplementary Fig. 7b, c), but not to the extent as in PC3 cells (Fig. 3b, d). Thus, PTEN-positive prostate cancer DU145 cells with deletion of *ARID4B* can still grow and form tumor (Supplementary Fig. 7b, c).

In human prostate cancers, we found that *ARID4B* is consistently retained in human prostate carcinoma cohorts with deep deletion of *PTEN* (Fig. 3a), suggesting the mutually exclusive trend of deletions between *ARID4B* and *PTEN*. However, not all human prostate cancers are driven by loss of *PTEN* or activation of the PI3K signaling. In PTEN-positive prostate cancers, function of the *ARID4B*-PI3K axis likely plays a less important role, and *ARID4B* may not be required for tumorigenesis. Therefore, deletion of *ARID4B* may occasionally occur in human PTEN-positive prostate cancer cases (0% - 3%) (Fig. 3a and Supplementary Fig. 1a, left).

In the Discussion section, we added the following paragraphs.

“In the PC3 (PTEN-deficient) cells, knockout of *ARID4B* efficiently inhibited cancer cell growth and tumor formation in xenografts, suggesting that PTEN-deficient prostate cancer cells depend on *ARID4B*. In the DU145 (PTEN-intact) cells, knockout of *ARID4B* also suppressed cell growth and tumor formation in xenografts, but not as much as in PC3 cells. PTEN-deficient PC3 cells may be more dependent on activation of the PI3K-AKT pathway for tumor growth than the PTEN-intact DU145 cells. This would explain why the AKT inhibitor MK-2206 inhibited cell growth more effectively in PC3 cells than DU145 cells (Supplementary Fig. 17a, b).

In PTEN-deficient PC3 cells, ablation of *PTEN* activates the PI3K/AKT pathway, leading to tumorigenesis (Supplementary Fig. 17c, left). In *ARID4B*KO PC3 cells, knockout of *ARID4B* compromises activation of the PI3K/AKT pathway, thus efficiently inhibiting tumorigenesis elicited by *PTEN* deficiency (Supplementary Fig. 17c, right). In PTEN-intact DU145 cells, *PTEN* dephosphorylates PIP3 into PIP2, which opposes the PI3K/AKT signaling pathway (Supplementary Fig. 17d, left), suggesting tumorigenesis of DU145 cells is less dependent on the PI3K-AKT signaling than PC3 cells. Therefore, knockout of *ARID4B* that inactivates the PI3K/AKT pathway only moderately suppressed tumorigenesis of DU145 cells (Supplementary Fig. 17d, right).”

Fig. S17

Q3) The authors should discuss why the predictive power of ARID4B alone (Figure S1d) appears more robust than the 3 gene signature in Figure 5i.

A3) Depending on the objective, different methods were chosen to calculate the predictive power in Supplementary Fig. 1d and Fig. 6d (Fig. 5i in the previous version of the manuscript).

In the Methods section of this revised manuscript, we had the following paragraphs.

“As shown in Supplementary Fig. 1d, Kaplan–Meier survival analysis was conducted using the PROGgeneV2 prognostics database⁴⁶ to seek correlations between expression of *ARID4B* in prostate tumors and recurrence-free survival in patients from the GSE40272²⁸ and the ref.²⁷ datasets. Levels of *ARID4B* expression were used to define high and low risk scores. The two groups (high and low expression of *ARID4B*) were generated by splitting patients with the median value of *ARID4B* expression.”

“In Fig. 6d, e and Supplementary Fig. 13b-e, we demonstrated the predictive power of the *PTEN/ARID4B/PIK3CA* three-gene signature for tumor recurrence in three datasets: TCGA provisional dataset, GSE21032, and GSE40272. The prognostic index (PI) was used to define the high and low risk scores, and the two risk groups (high and low) were generated by splitting patients with the median PI. The PI, as the linear component of the Cox model, was calculated as $PI = b_1x_1 + b_2x_2 + \dots + b_px_p$, where x is the gene expression value and the b is a risk coefficient obtained from the Cox fitting. The fitting was performed in R (<http://cran.r-project.org>) using the survival package.

The multivariate Cox proportional hazards model of the *ARID4B/PTEN/PIK3CA* signature combined expression levels of these three genes with other clinical and pathological variables into an integrated risk score model to predict time to recurrence in prostate cancer patients from the TCGA provisional dataset using statistical software SPSS19.0 and from two datasets GSE21032² and GSE40272²⁸ using the SurvExpress web tool⁴⁷ (<http://bioinformatica.mty.itesm.mx:8080/Biomatec/SurvivaX.jsp>). Briefly, the PI was used to define and generate high and low risk groups with classical multivariate Cox model, with higher PI representing higher risk. Groups at high and low risk for recurrence were generated by splitting values at the median PI. In addition, Kaplan-Meier survival curves were used to show that the *ARID4B/PIK3CA/PTEN* gene could distinguish between the two groups for the prostate cancer recurrence. Patients in high-risk groups had a lower survival rate than those in the low-risk group in the TCGA provisional, GSE21032 ($P = 0.002$), and GSE40272 ($P = 0.006$) datasets.”

Q4) The origin, growth conditions and verification of the cell lines used in Supplementary Studies-LNCaP, C4-2B, RWPE-1, PZ-HP-7 and HPrEC- were not included in the manuscript.

A4) In the Methods section of this revised manuscript, we added the following information.

“All cell lines were obtained from ATCC. PC3 and DU145 cells were cultured in F-12K Medium and Eagle's Minimum Essential Medium supplemented with fetal bovine serum (FBS) to a final concentration of 10%, respectively. C4-2B cells were cultured in Dulbecco's modified Eagle's

medium (DMEM)/F12 (4:1) supplemented with heat-inactivated FBS (to 10%) and T-medium supplement (to 1×). To prepare 10× T-medium supplement in 100 ml of 0.1% BSA (Sigma cat# A8022) in PBS, the following components were added: 50 mg insulin (Gibco cat# 12585-014), 136 ng triiodo-L-thyronine (Sigma cat# T2877), 50 mg transferrin (Sigma cat# T4382), 2.5 mg D-Biotin (Sigma cat# 47868), and 250 mg adenine (Sigma cat# A3159). RWPE-1 and PZ-HPV-7 cells were maintained in Keratinocyte Serum Free Medium (K-SFM, GIBCO 17005-042), supplemented with 0.05 mg/ml of bovine pituitary extract and 5 ng/ml of human recombinant epidermal growth factor. HPrEC cells were maintained in prostate epithelial cell basal medium (ATCC, PCS-440-030) supplemented with prostate epithelial cell growth kit (ATCC, PCS-440-040).”

In Supplementary Fig. 1c of this revised manuscript, we added the following information.

Cell line	Tumorigenicity	PTEN	Androgen sensitivity
PC3	High	-	Insensitive
DU145	Moderate	+	Insensitive
LNCaP	Low	-	Sensitive
C4-2B	High	-	Insensitive
RWPE-1	No	+	Sensitive
PZ-HPV-7	No	+	Sensitive
HPrEC	No	+	Sensitive

Q5) Have the authors validated the gene expression studies by IHC of primary human prostate tumors?

A5) As requested, we performed immunohistochemical staining on prostate cancer tissue microarrays (TMAs) to investigate the clinical relevance of the newly identified PTEN-ARID4B-PI3K axis in human prostate cancer. Results support our findings, and suggest a negative correlation between ARID4B and PTEN expression levels in human prostate adenocarcinoma (Fig. 6b, c and Supplementary Fig. 13a) and a positive correlation between ARID4B and p110 α levels in human prostate adenocarcinoma (Fig. 6b, c and Supplementary Fig. 13a).

In the Results section, we added the following paragraph.

“To investigate the clinical relevance of the newly identified PTEN-ARID4B-PI3K axis in human prostate cancer, we performed immunohistochemical staining with antibodies against ARID4B, PTEN, and p110 α on tissue microarrays (TMAs) consisting of 118 prostate adenocarcinoma samples from 85 patients and 73 prostate hyperplasia samples from 36 patients. Results revealed that ARID4B levels were higher in samples with prostate adenocarcinoma compared to prostate hyperplasia (Fig. 6a, c, left and Supplementary Fig. 13a, left), consistent with results from Oncomine database analysis showing increased expression of *ARID4B* in human prostate carcinoma compared to normal prostate glands (Supplementary Fig. 1b). In addition, levels of PTEN were diminished in prostate adenocarcinoma (Fig. 6a, c, middle and Supplementary Fig.

13a, middle), whereas the levels of p110 α were elevated (Fig. 6a, c, right and Supplementary Fig. 13a, right). Furthermore, expression of ARID4B and PTEN were negatively correlated in human prostate adenocarcinoma (Fig. 6b, c and Supplementary Fig. 13a), whereas ARID4B and p110 α levels were positively correlated in human prostate adenocarcinoma (Fig. 6b, c and Supplementary Fig. 13a).”

Minor Criticisms:

Q1) The manuscript contains numerous grammatical errors that the authors should have addressed before submitting a revised manuscript.

A1) This revised manuscript has been professionally edited.

REVIEWERS' COMMENTS:

Reviewer #1 (Remarks to the Author):

The authors have significantly revised the manuscript, which now importantly includes protein analysis of PTEN, ARID4B and p110alpha using a tissue microarray. These findings strengthen their three-gene signature claim, and I believe would be of interest to the readership of Nature communications.

Minor points:

1. Line 242-243: Sentence meaning is unclear and should be re-phrased, e.g. "PTEN lipid phosphatase activity dephosphorylates Phosphatidylinositol (3,4,5)-trisphosphate (PIP3) back into phosphatidylinositol 4,5-bisphosphate (PIP2) to negatively regulate the PI3K signaling cascade. Thus, PTEN loss promotes PI3K-AKT signaling⁶."
2. Line 243-244; Sentence should be amended to improve accuracy: To demonstrate that deficiency of PTEN correlates with increased expression of ARID4B via activation of the PI3K-AKT pathway
3. To further clarify the low/high risk groups for the PTEN/ARID4B/PIK3CA three-gene signature, the authors should provide further definition of the risk groups throughout the manuscript. For instance, does the high-risk group represent PTEN^{low}/ARID4B^{high}/PIK3CA^{high} mRNA expression?
4. Line 259-268: To improve clarity, the authors should avoid describing the IHC staining analysis as protein "levels" and instead refer to the scoring methodology (i.e. no. of positive epithelial cells).
5. To ensure reproducibility, the authors should provide the dilutions and/or concentrations of antibodies used for IHC and briefly describe the TMA IHC staining protocol. Is the PRC121 TMA the PRC1021 TMA?

Reviewer #2 (Remarks to the Author):

The authors have done an excellent job of addressing the concerns of the previous reviewers. The manuscript presents a convincing study demonstrating a role for the ARID4B protein in regulating the PI3K signaling pathway in the development of PTEN-deficient prostate cancer. This paradigm may lead to new therapeutic approaches for this disease through the targeting of ARID4B activities. The strength of the studies comes from the genetically-engineered mouse model where deletion of ARID4B clearly inhibits development of PTEN-deficient prostate cancer. The analysis of the human tumor data and the manipulation of cell lines in culture support the mouse studies as well as the pathway model proposed by the authors. This novel signaling pathway should impact the studies of other human cancers beside prostate.

Minor concern- Despite the "professional editing" of the revised manuscript, grammatical errors still appear.

Re: Final revisions for manuscript NCOMMS-18-28842B

REVIEWERS' COMMENTS:

Reviewer #1 (Remarks to the Author):

The authors have significantly revised the manuscript, which now importantly includes protein analysis of PTEN, ARID4B and p110alpha using a tissue microarray. These findings strengthen their three-gene signature claim, and I believe would be of interest to the readership of Nature communications.

Minor points:

Q1. Line 242-243: Sentence meaning is unclear and should be re-phrased, e.g. “PTEN lipid phosphatase activity dephosphorylates Phosphatidylinositol (3,4,5)-trisphosphate (PIP3) back into phosphatidylinositol 4,5-bisphosphate (PIP2) to negatively regulate the PI3K signaling cascade. Thus, PTEN loss promotes PI3K-AKT signaling⁶.”

A1: As suggested, in this second revised manuscript, the sentence (Line 242-243 in the previous version) was changed to the following.

“PTEN lipid phosphatase activity dephosphorylates PIP3 into PIP2 to negatively regulate the PI3K signaling cascade. Thus, loss of PTEN promotes PI3K-AKT signaling^{6, 7}.”

Q2. Line 243-244; Sentence should be amended to improve accuracy: To demonstrate that deficiency of PTEN correlates with increased expression of ARID4B via activation of the PI3K-AKT pathway.

A1: As suggested, in this second revised manuscript, the sentence (Line 243-244 in the previous version) was changed to the following.

“To demonstrate that PTEN deficiency correlates with increased ARID4B expression *via* activation of the PI3K-AKT pathway, we showed that knockdown of *AKT* in the PTEN-deficient prostate cell line PC3 decreased expression of ARID4B (Fig. 5h).”

Q3. To further clarify the low/high risk groups for the PTEN/ARID4B/PIK3CA three-gene signature, the authors should provide further definition of the risk groups throughout the manuscript. For instance, does the high-risk group represent PTEN^{low}/ARID4B^{high}/PIK3CA^{high} mRNA expression?

A1: As suggested, in the Methods section of this second revised manuscript, we provided further definition of the risk groups as following.

“The predictive power of the *PTEN/ARID4B/PIK3CA* three-gene signature for tumor recurrence was calculated from three datasets: the TCGA provisional, the GSE21032², and the GSE40272²⁸ datasets. The prognostic index (PI) was used to define high- and low-risk scores, and the two risk groups (high and low) were generated by splitting patients’ data at the median PI. The PI of multiple genes was calculated as $PI = x_1\beta_1 + x_2\beta_2 + \dots + x_p\beta_p$, where x is the gene expression value and β is a risk coefficient obtained from the Cox fitting. The fitting was performed in R (<http://cran.r-project.org>) using the survival package. The β represents the correlation between gene expression levels and the number of events (in our case, death from cancer recurrence). A positive β means a higher gene expression level may contribute to a higher death rate, and a negative β means a higher gene expression value may contribute to a lower death rate. In our case, *PTEN* has a negative β coefficient, while *ARID4B* and *PIK3CA* both have positive β coefficients. Patients with lower *PTEN* expression levels or with higher *ARID4B* and/or *PIK3CA* levels would have higher calculated PIs, and therefore would be classified into the high-risk group.

The multivariate Cox proportional hazards model of the *ARID4B/PTEN/PIK3CA* signature combined expression levels of these three genes with time to recurrence in prostate cancer patients from the TCGA provisional dataset using statistical software SPSS19.0, and from the GSE21032² and GSE40272²⁸ datasets using the SurvExpress web tool⁴⁷ (<http://bioinformatica.mty.itesm.mx:8080/Biomatec/SurvivaX.jsp>). In addition, Kaplan-Meier survival curves were used to test whether the *ARID4B/PIK3CA/PTEN* signature distinguishes between high- and low-risk groups for the prostate cancer recurrence in the TCGA provisional ($P = 0.025$), GSE21032² ($P = 0.002$), and GSE40272²⁸ ($P = 0.006$) datasets. Statistical analysis: log-rank test.”

Q4. Line 259-268: To improve clarity, the authors should avoid describing the IHC staining analysis as protein “levels” and instead refer to the scoring methodology (i.e. no. of positive epithelial cells).

A1: As suggested, in this second revised manuscript, the sentence (Line 259-268 in the previous version) was changed to the following.

“Results revealed that expression of ARID4B was higher in prostate adenocarcinoma compared to prostate hyperplasia (Fig. 6a, left), shown as an increase in the number of ARID4B positive cells in prostate adenocarcinoma samples (Fig. 6c, left and Supplementary Fig. 11a, left).”

Q5. To ensure reproducibility, the authors should provide the dilutions and/or concentrations of antibodies used for IHC and briefly describe the TMA IHC staining protocol. Is the PRC121 TMA the PRC1021 TMA?

A1: In this second revised manuscript, we corrected the typo PRC121 TMA to the PRC1021 TMA.

As suggested, we provided the dilutions of antibodies used for IHC staining and described the TMA IHC staining protocol in the Methods section of this second revised manuscript as appears below.

“Immunohistochemical staining of two TMAs (PRC1021 and PRD961) and image scanning services were provided by Pantomics (Pantomics, Inc. USA). Two TMA slides, PRC1021 and PRD961, included 118 prostate adenocarcinoma cores from 85 patients, 73 prostate hyperplasia cores from 36 patients, and 5 benign tumor cores. These TMA slides were baked at 60°C for 30 minutes and then deparaffinized. After antigen retrieval, slides were incubated with primary antibodies: anti-ARID4B (A302-233A, Bethyl Laboratories; dilution 1: 400), anti-PTEN (A2B1; sc-7974, Santa Cruz Biotechnology; dilution 1: 100), and anti-p110 α (C73F8; 4249, Cell Signaling Technology; dilution 1: 200). Slides were washed with 0.1% Triton X-100/PBS buffer, and incubated with the secondary antibody conjugated with horseradish peroxidase. Signal detection was carried out using DAB (3,3'-diaminobenzidine) substrate. Slides were counterstained with hematoxylin. The scanned images were analyzed with QuPath (v0.1.2)⁴⁰. Expression of ARID4B, PTEN, and p110 α is presented as the number of positive epithelial cells per mm².”

Reviewer #2 (Remarks to the Author):

The authors have done an excellent job of addressing the concerns of the previous reviewers. The manuscript presents a convincing study demonstrating a role for the ARID4B protein in regulating the PI3K signaling pathway in the development of PTEN-deficient prostate cancer. This paradigm may lead to new therapeutic approaches for this disease through the targeting of ARID4B activities. The strength of the studies comes from the genetically-engineered mouse model where deletion of ARID4B clearly inhibits development of PTEN-deficient prostate cancer. The analysis of the human tumor data and the manipulation of cell lines in culture support the mouse studies as well as the pathway model proposed by the authors. This novel signaling pathway should impact the studies of other human cancers beside prostate.

Q1: Minor concern- Despite the "professional editing" of the revised manuscript, grammatical errors still appear.

A1: We have used the English language editing service to edit this second revised manuscript.